# Modularity and composite diversity affect the collective gathering of information online

Niccolò Pescetelli [1,2 ✉], Alex Rutherford[1,2] & Iyad Rahwan [1,2]

Many modern interactions happen in a digital space, where automated recommendations and homophily can shape the composition of groups interacting together and the knowledge that groups are able to tap into when operating online. Digital interactions are also characterized by different scales, from small interest groups to large online communities. Here, we manipulate the composition of groups based on a large multi-trait profiling space (including demographic, professional, psychological and relational variables) to explore the causal link between group composition and performance as a function of group size. We asked volunteers to search news online under time pressure and measured individual and group performance in forecasting real geo-political events. Our manipulation affected the correlation of forecasts made by people after online searches. Group composition interacted with group size so that composite diversity benefited individual and group performance proportionally to group size. Aggregating opinions of modular crowds composed of small independent groups achieved better forecasts than aggregating a similar number of forecasts from non-modular ones. Finally, we show differences existing among groups in terms of disagreement, speed of convergence to consensus forecasts and within-group variability in performance. The present work sheds light on the mechanisms underlying effective online information gathering in digital environments.

[1] Center for Humans and Machines, Max Planck Institute for Human Development, Berlin, Germany. [2] Media Lab, Massachusetts Institute of Technology, Cambridge, MA, USA. ✉email: Pescetelli@mpib-berlin.mpg.de

Understanding how people collect information about world events and discuss this knowledge with others online to form shared opinions is a crucial and timely research question. In the past decade, there have been widespread concerns that search engines and news filtering algorithms may contribute to the formation of clusters of individuals with highly correlated information and poorly diversified news sources[1–3]. Little is known about the exact mechanisms underlying algorithmic personalization, but content is often provided by clustering users on highly dimensional feature spaces, along shared variables (demographics, geo-location, social network, tastes, and past behavior)[4–8]. Furthermore, people sharing traits are more likely to voluntarily cluster together in online communities, a phenomenon known as homophily[9,10]. One question is whether recommendation algorithms and homophily can impact the ability of online groups to collectively search and use online information to form accurate representations of future events, especially under high time pressure and uncertainty—namely when the opportunities for rational debates are scarce[11,12].

In this paper, we manipulate the size/modularity of online groups and their composition along a heterogeneous profiling space (including demographic, professional, political, relational, and psychological features, see Supplementary Information §1 and 2). Both factors are expected to affect the amount and independence of information that a group can tap into[13–15]. We tested people's ability to collectively retrieve task-relevant information online to form accurate representation of the world. We measure individual and group performance as Brier errors in forecasting real geo-political events (Supplementary Table 1). This task has high ecological validity. Forecasting problems were independently selected as part of a national forecasting tournament and were representative of challenges commonly facing experts and professional intelligence analysts. These forecasting problems are characterized by high degrees of uncertainty and correlated information between respondents, dependence on multiple indicators (e.g., economics, politics, social unrest, etc.), and, importantly, time criticality (i.e.,there are huge costs associated with making the correct prediction too late). Importantly, the difficulty and specificity of the forecasting problems ensured that individual and group forecasts in our study were driven by the information participants could retrieve online in a short amount of time, rather than domain-specific information already possessed by the participant before the experiment. General knowledge questions or more familiar forecasting problems would have confounded participants' prior knowledge and information they retrieved online. For this reason, we expected our composite measure of diversity to affect a group's ability to effectively search relevant information online.

Diversity is a highly heterogeneous construct touching several disciplines[16–19]. From an informational standpoint, psychologists have recognized the importance of group diversity for information independence, group performance, resilience to group biases, complex thinking, creativity, and exploration of large solution spaces[20–30]. The approach used in psychology is aimed at studying single dimensions of diversity (e.g., skill, age, and race[25,30,31]). Contrary to this, we are here interested in the effects that sorting people based on a large multi-trait space (Fig. 1A) can have on the information diversity that a group can forage online. During the pretest phase, we surveyed participants along 29 dimensions (see Supplementary Information for a full list of features considered). Each participant represented a data point along this profiling space based on their responses to the survey. Questions included demographic indicators (such as age, sex and education, and race), professional indicators (e.g., hours a day spent working with things, ideas, people, or data), political preferences (left/right-wing), geographic indicators (e.g., countries

visited in the past 6 months), as well as relational variables (e.g., political orientation of your average friend), and cognitive indicators (e.g., cognitive reflective test[32]). Many of the features used in this study—such as demographics, political orientation, and personality traits—can be easily inferred from digital traces, and used to customize searches and recommend content[33–36]. Although some of these features (like demographics) are known to psychologists not to affect information diversity per se in an offline setting[37,38], they may do so in an online environment that maps interindividual differences into information access. Arguably, the more distant two people are on an arbitrarily large profiling space, the less likely it is they belong to the same online information bubble. Given the difficulty of disentangling the causal contributions of group composition on performance, we here employ an experimental design[23,25,26,30] to create groups of people who were close or distant to each other along our multi-trait profiling space. To manipulate our composite measure of diversity, we used a data-driven clustering algorithm (DBSCAN) that segmented participants based on their Euclidean distance on the profiling space. Half of the participants (core segment) corresponding to the center of the distribution was randomly assigned to interact with the rest 25% most similar (inner segment) or 25% most dissimilar (outer segment) individuals in the sample (Fig. 1B, C). Euclidean distance was strongly correlated with standard deviance, another popular measure of diversity with multidimensional input ($r$:0.92, $p < 0.001$; Supplementary Fig. 13), suggesting robustness across alternative measures.

As the scale of online collaboration widely varies (from small interest groups to large online communities), we wanted to characterize the effects of group composition as a function of group size. Orthogonally to diversity, we randomized the size and modularity of the online collective. Manipulating group size or the number of groups interrogated can have positive effects on group performance, by reducing error cascades[14,15,39–42]. Smaller groups are more likely to maximize accuracy in environments characterized by inter-judgment correlations thanks to their inherent noise and greater exploratory behavior[41,43–47]. Furthermore, aggregating information from multiple smaller interacting groups performs better than traditional wisdom-of-crowd because it insulates the aggregate from correlated errors[14]. In other words, rather than interrogating one single large crowd ($M = 1$), greater accuracy is obtained by dividing the large crowd into smaller, but independent (i.e., noncommunicating) groups ($M > 1$). We call this feature modularity. Modularity maintains information diversity (across groups) in spite of herding (within groups). However, prior studies[14] were performed on estimation tasks, where crowds are known to perform well[48]. Whether the same results generalize to more complex real-world problems are unknown.

After sorting people into groups of different sizes and composition, participants were asked to give for each forecasting problem an initial guess (initial forecast). Then they were asked to revise it after privately browsing online (revised forecast), and after debating with others online (private final forecast and group consensus forecast). A preregistration of our hypotheses is available via OSF. At the individual level, we expected alignment of opinions and improved accuracy due to online browsing and social influence. At the aggregate level, we expected group diversity and modularity to positively affect aggregate performance. No predictions were made regarding the direction of their interaction. Exact analyses were not preregistered. Aggregation followed the same procedure described in ref. [14]. Small groups (~5 people) were approximately the square root of large groups (~25 people; cf. ref. [15]).

Our findings show that the closer (more similar) individuals were on the profiling space the more correlated their forecasts

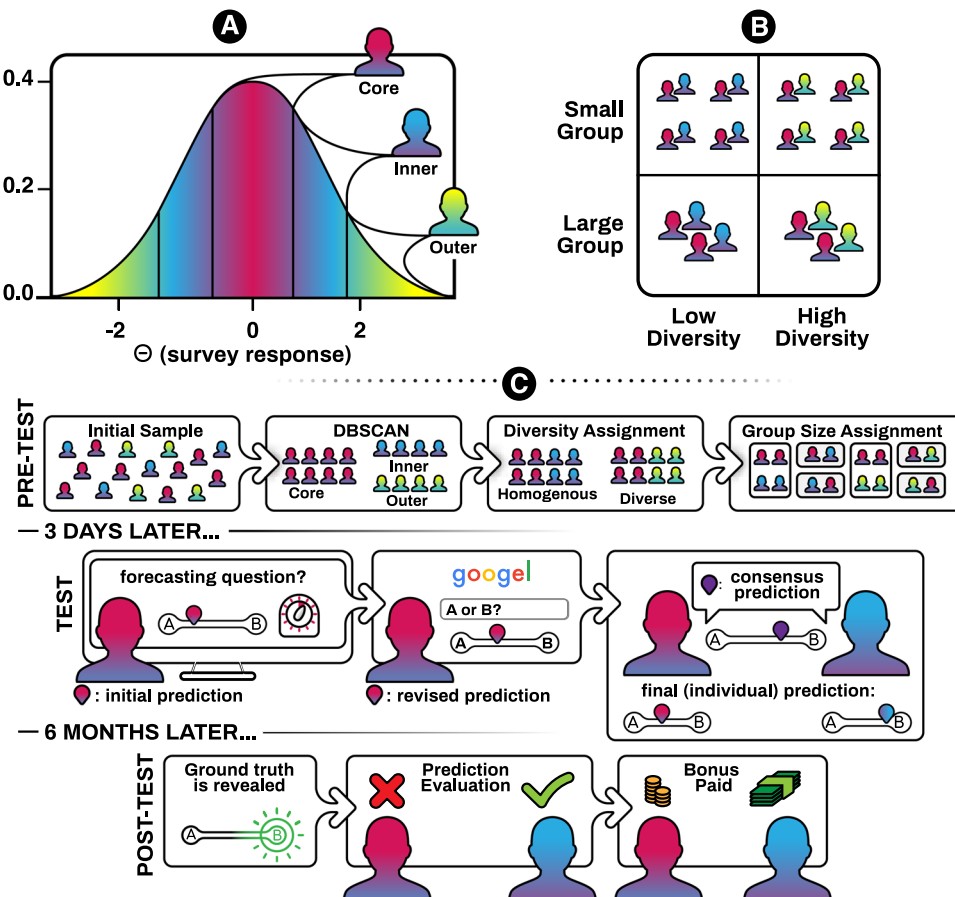

**Fig. 1 Experimental design. A** One-dimensional representation of the partitioning of the $\Theta$ space by the DBSCAN algorithm. In reality, $\Theta \in \mathbb{R}^D$, where $D$ is the number of dimensions considered ($D = 29$). **B** 2 × 2 design with factors: diversity (low vs. high) × modularity (low vs. high). Low vs. high diversity manipulation was achieved by matching the core participants to either the inner segment participants (low diversity condition) or the outer segment (high diversity condition). **C** Experimental procedure. At pretest time (upper row), participants were administered a battery of surveys that were used to cluster them into a core, inner, and outer segments (DBSCAN). Core participants were then randomized to a diversity and modularity condition. At test time, they answered eight forecasting problems first alone (stages 1 and 2) and then within their groups (stage 3).

became after online searches. Group diversity benefited individual and aggregated performance and interacted with group size so that large groups benefited from it more than smaller ones. Analysis of forecasts distributions and exploratory linguistic analysis of chat data showed slower consensus building, greater disagreement, and greater variance in group members' performance impacting large groups with higher composite diversity score less negatively than small ones. We also find that forcing individuals to reach a consensus as opposed to simply being exposed to social information benefits their ability to forecast future events. These findings inform how social interaction online can affect real-life problem solving in complex information environments. We discuss these results in light of the recent literature on collective behavior in ecology and social science.

## Results

**Multidimensional profiling**. Exploratory analyses were ran to characterize our composite diversity measure. Trait diversity correlated with information diversity only after (but not prior) online browsing. After browsing, larger Euclidean distance along the profiling space $\Theta$ between pairs of individuals was inversely related to the correlation coefficient of the forecasts made by the same two individuals (initial: $r = 0.12$ $p = 0.38$; revised: $r = -0.39$, $p = 0.006$; final: $-0.056$, $p < 0.001$). This indicates that online browsing produced greater alignment of beliefs proportionally to individual similarity.

A principal component analysis was ran to characterize post hoc the participants' response distribution during the pretest phase. Trait variation in our population was highly structured, about five components explained ~90% of the variance (Supplementary Fig. 13), suggesting most trait dimensions were redundant or showed little variation. Principal components correlated with ethnic-cultural and socio-political variability in our sample (Supplementary Figs. 14–16). A parallel analysis (Supplementary Fig. 17) showed eight principal components, reported in Supplementary Information. Participants segmentation into core, inner, and outer segments was already visible on a low-dimensional principal component projection (Supplementary Fig. 18), confirming that core participants were more similar (along the principal components) to participants belonging to the inner segment than to participants belonging to the outer segment. Finally, we checked that no principal component was trivially related with opinion diversity or performance (Supplementary Figs. 22 and 23).

**Individual-level performance**. For each forecast, a Brier error score (range [0, 2]) was computed according to Eq. (1) in the "Methods" section. Distributions of individual and aggregated errors are reported in Supplementary Fig. 2. Errors were larger (worse performance) for initial ($\beta = 0.62$, SE = 0.09, $t = 6.88$, $p < 5.81e - 12$), revised ($\beta = 0.69$, SE = 0.08, $t = 7.77$, $p < 7.73e - 15$), and final ($\beta = 0.23$, SE = 0.09, $t = 2.39$, $p = 0.01$) forecasts

compared to consensus forecasts (Fig. 2A), indicating an overall forecast improvement over repeated judgments (Table 1A and Supplementary Table 4). Against our preregistered hypotheses, initial forecasts were numerically, but nonsignificantly better than revised forecasts. Both initial and revised forecasts however were worse than following forecasts ($\beta$s < −0.38, SEs < 0.09, $t$s < −5.12, $p$s < $2.94e − 07$), confirming our preregistered hypothesis of an accuracy improvement due to social interaction[49]. Final and consensus forecasts contained the same socially acquired information and were made in random order. Surprisingly, errors were smaller for the consensus than the final forecast, suggesting that forcing consensus (rather than simple social exposure) improved individual forecasting accuracy.

We conducted an exploratory analysis on the effects that our composite measure of diversity (reference: lower composite diversity score) and group size (reference: large groups) assignment had on individual forecasting accuracy (Table 1B and Supplementary Table 5). Initial and revised forecasts were not affected by our manipulation and were thus excluded from this analysis. Notice that at the individual level, we can only test whether interacting in small or larger groups has an effect on forecasting error, given that modularity is a group-level feature (see Supplementary Information §2). A model with an interaction term was superior to one without, notwithstanding the added complexity (d.f. = 8, $\chi^2$ = 7.63, $\chi^2$d.f. = 1, $p$ = 0.005). Working in groups with higher composite diversity score marginally predicted better individual performance ($\beta$ = −0.37, SE = 0.20, $t$ = −1.83, $p$ = 0.06). Participants in homogeneous small groups performed nonsignificantly worse their counterparts in homogeneous larger groups ($\beta$ = −0.20, SE = 0.20, $t$ = −0.99, $p$ = 0.31). The beneficial effect of composite diversity on individual performance was positively affected by group size, suggesting that individual interaction with diverse peers was more beneficial in large than small groups ($\beta$ = 0.82, SE = 0.29, $t$ = 2.85, $p$ = 0.004; Fig. 2B). The same interaction was found when using average profiling distance (continuous) rather than diversity treatment (categorical) as a measure of diversity (Supplementary Table 6 and Supplementary Fig. 3). Similar conclusions were reached when limiting our analysis to final forecasts only, but not to consensus forecasts only (Supplementary Tables 7–8), suggesting that these results were likely driven by final individual beliefs rather than collective ones.

**Group-level performance**. In forecasting like in democratic decisions, aggregated individual judgments are more informative than individual ones. At the aggregate level, we can now ask whether modularity and hierarchical aggregation can improve forecasting accuracy in our online information gathering task[14,15]. For each group, we computed an aggregate forecast by taking the median forecast in the group for each forecast type. By definition, we have only one group per diversity treatment in the non-modular condition ($M = 1$), but multiple subgroups in the modular condition ($M > 1$). Thus, aggregating judgments in the high modularity condition proceeded by aggregating forecasts in each group first, and then aggregating aggregates[14]. An exploratory analysis on aggregate forecasts, showed that consensus forecasting errors were lower than both initial ($\beta$ = 0.68, SE = 0.22, $t$ = 2.97, $p$ = 0.002) and revised ($\beta$ = 0.59, SE = 0.23, $t$ = 2.60, $p$ = 0.009) errors, suggesting a benefit of social interaction (Table 1C and Supplementary Table 9). The advantage of consensus over final forecasts disappeared at the aggregate level ($\beta$ = −0.12, SE = 0.29, $t$ = −0.43, $p$ = 0.66; Fig. 3A).

Our main hypotheses consisted in analyzing the effect of group assignment on aggregated forecasting errors during the social exchange. A model with fixed effects for composite diversity,

modularity and an interaction between the two provided better fit than one without interaction (d.f. = 7, $\chi^2$ = 6.10, $p$ = 0.01). As predicted, aggregate forecasts from groups higher on composite diversity were better than aggregate forecasts from homogeneous groups ($\beta$ = −0.56, SE = 0.23, $t$ = −2.39, $p$ = 0.01; baseline: large, Table 1D and Supplementary Table 10). Also as predicted, aggregated forecasts obtained from smaller/modular groups were better than from larger/non-modular groups ($\beta$ = −0.82, SE = 0.26, $t$ = −3.10, $p$ = 0.001; baseline: homogeneous). Finally, we found an interaction between composite diversity and modularity whose direction we did not predict ($\beta$ = 0.93, SE = 0.38, $t$ = 2.43, $p$ = 0.01), indicating that the beneficial effect of composite diversity on aggregate forecasting accuracy was significantly greater in large groups over smaller groups (Fig. 3B).

**Disagreement, consensus reaching, and performance variability**. To understand why diversity interacted with group size, we performed three main exploratory analyses. First, we analyzed the distribution of forecasts produced by each group in different questions (Supplementary Fig. 2). In particular, we were interested in the disagreement between participants' estimates (diversity of opinions in ref. [50]), namely the dispersion (standard deviation) of the forecast distribution within a group. A greater standard deviation suggests more conflicting views and thus more conflicting evidence for the group to resolve, when trying to reach a consensus under time pressure. Compared to initial forecasts, disagreement was lower in final forecasts ($\beta$ = −4.41, SE = 1.18, $t$ = −3.72, $p$ < 0.001) and higher in revised forecasts ($\beta$ = 5.06, SE = 1.18, $t$ = 4.27, $p$ < 0.001), suggesting (surprisingly) an increase in the spread of opinions after online information search and (unsurprisingly) opinion alignment after social interaction (Supplementary Table 15). We found no main effects of diversity ($\beta$ = −0.48, SE = 2.36, $t$ = −0.20, $p$ > 0.8) or group size ($\beta$ = −3.51, SE = 1.80, $t$ = −1.94, $p$ > 0.05). However, diversity interacted with group size suggesting that it had a smaller effect on disagreement in large groups compared to small ones ($\beta$ = 7.11, SE = 2.60, $t$ = 2.73, $p$ = 0.006). Residual disagreement remained even after people had the chance to come to a consensus, as observed in final forecasts (Fig. 4A).

Our second analysis suggests that online information gathering affected within-group variability in performance. Larger performance variability indicates that a group contains members who are very accurate (on average across the eight individual forecasting problems (IFPs)) and members who are quite poor. Performance variability is typically associated with reduced collective intelligence[51,52]. In the initial stage people's accuracy was similar to each other (~0.1–0.2 standard deviations of Brier scores), but variability increased in small diverse groups after online information gathering. This effect was not as nearly as pronounced for small homogeneous groups and large groups (Fig. 4B), suggesting that browsing selectively negatively impacted small groups scoring higher on our composite diversity measure.

A third factor we investigated was whether our manipulation affected the process of consensus reaching through online deliberation (see Supplementary Information §5 and 6). We manually labeled forecast estimates mentioned by participants during the deliberation phase and fitted a model representing convergence of these estimates to the consensus forecast. Group composite diversity decreased consensus reaching times ($\beta$ = −0.31, SE = 0.12, $t$ = −2.55, $p$ = 0.01, baseline: large). Also small groups showed quicker consensus reaching than large ones ($\beta$ = −0.46, SE = 0.10, $t$ = −4.68, $p$ < 0.001, baseline: homogeneous; Supplementary Table 16). A positive interaction between the two factors indicated that speed in consensus reaching observed in diverse

**Table 1 Generalized mixed-effects models on individual and aggregated errors.**

| Effect | Estimate | Fitted Brier score | SE | t | p |
|---|---|---|---|---|---|
| **(A) Individual forecasting error as a function of forecast type** | | | | | |
| Intercept | −2.14224 | 0.1173915 | 0.24230 | −8.841 | <**2e − 16** |
| Initial | 0.62237 | 0.2187395 | 0.09040 | 6.884 | **5.81e − 12** |
| Revised | 0.69532 | 0.2352946 | 0.08947 | 7.772 | **7.73e − 15** |
| Final | 0.23849 | 0.1490093 | 0.09979 | 2.390 | **0.0169** |
| **(B) Individual forecasting error as a function of diversity and group size** | | | | | |
| Intercept | −1.96631 | 0.139972 | 0.30877 | −6.368 | **1.91e − 10** |
| Final | 0.20997 | 0.1726759 | 0.07814 | 2.687 | **0.00720** |
| Diverse | −0.37285 | 0.1189339 | 0.20278 | −1.839 | 0.06595 |
| Small | −0.20011 | 0.1413602 | 0.20094 | −0.996 | 0.31932 |
| Diverse:small | 0.82956 | 0.2231896 | 0.29025 | 2.858 | **0.00426** |
| **(C) Aggregated forecasting error as a function of forecast type** | | | | | |
| Intercept | −1.8387 | 0.1590198 | 0.2508 | −7.331 | **2.29e − 13** |
| Initial | 0.6815 | 0.3143683 | 0.2293 | 2.972 | **0.00296** |
| Revised | 0.5999 | 0.2897372 | 0.2301 | 2.607 | **0.00913** |
| Final | −0.1281 | 0.1398955 | 0.2964 | −0.432 | 0.66557 |
| **(D) Aggregated forecasting error as a function of Diversity and Modularity** | | | | | |
| Intercept | −1.76627 | 0.1709691 | 0.33428 | −5.284 | **1.27e − 07** |
| Final | −0.06877 | 0.15960632 | 0.15360 | −0.448 | 0.65434 |
| Diverse | −0.56382 | 0.09082084 | 0.23514 | −2.398 | **0.01649** |
| Modular | −0.82268 | 0.07010727 | 0.26515 | −3.103 | **0.00192** |
| Diverse:modular | 0.93267 | 0.10137943 | 0.38254 | 2.438 | **0.01477** |

Table of analysis on forecasting errors (in Brier scores) for individual (A–B) and aggregated measures (C–D), and as a function of forecast type (A–C) and condition (B–D). Baselines for each factor: consensus, homogeneous, large/non-modular. The effect of final forecasts on individual errors (A) and the effects of composite diversity and the interaction between composite diversity and modularity (D) did not survive a Bonferroni correction. Boldface: $p < 0.05$; italics: $p < 0.10$. Tables B–C represent exploratory analyses. Hypotheses in tables A and D were preregistered. All analyses were also repeated with binarized accuracy (Supplementary Tables 11–14) and logit link function (Supplementary Table 15). For convenience, all tests refer to two-sided hypotheses and were calculated with the lmerTest package in R[63].

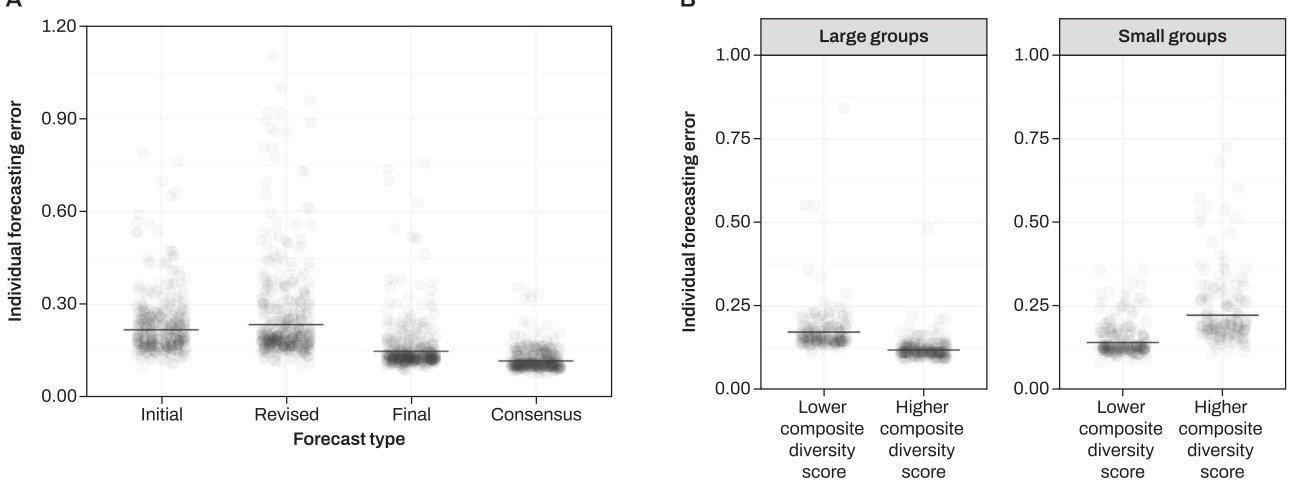

**Fig. 2 Individual-level analysis. A** Partial residuals plot showing the effect of forecasting type on individual forecasting error (measured in Brier scores). Lower numbers represent higher accuracy. Solid lines represent model fit. **B** Partial residuals plot showing the effect of diversity and group size on individual forecasting error (expressed in Brier scores). Solid lines represent model fit. Notice that, for visualization purposes, the graphs have been plotted onto the original error scale rather than log scale as in the fitted GLMM. Thus, large residuals should not cause concern[64]. See Supplementary Fig. 10 when using a logit link. Source data are provided as a Source data file.

groups decreased as a function of smaller group size ($\beta = 0.69$, SE = 0.17, $t = 4.004$, $p < 0.001$; Supplementary Figs. 6 and 7).

## Discussion

In this study, we experimentally manipulated the size and group composition of groups collaboratively gathering information online. We found that sorting groups based on a composite measure of diversity—including demographic, relational, and cognitive indicators—affected the correlation of beliefs of people only after they were asked to gather information online. Both social interaction and the need to reach an internal consensus via

deliberation improved people's forecasting accuracy. Collaborating in groups with higher composite diversity was beneficial for people's individual ability to forecast the future, proportionally to group size (Fig. 2). When aggregating judgments together using a simple median, this translated into an advantage of modular groups and groups with higher composite diversity, and an interaction between composite diversity and modularity (Fig. 3). We explored the mechanisms underlying this interaction with a range of exploratory analyses (Fig. 4).

The widespread use of automated content recommendation paired with people's tendency to interact with others who share

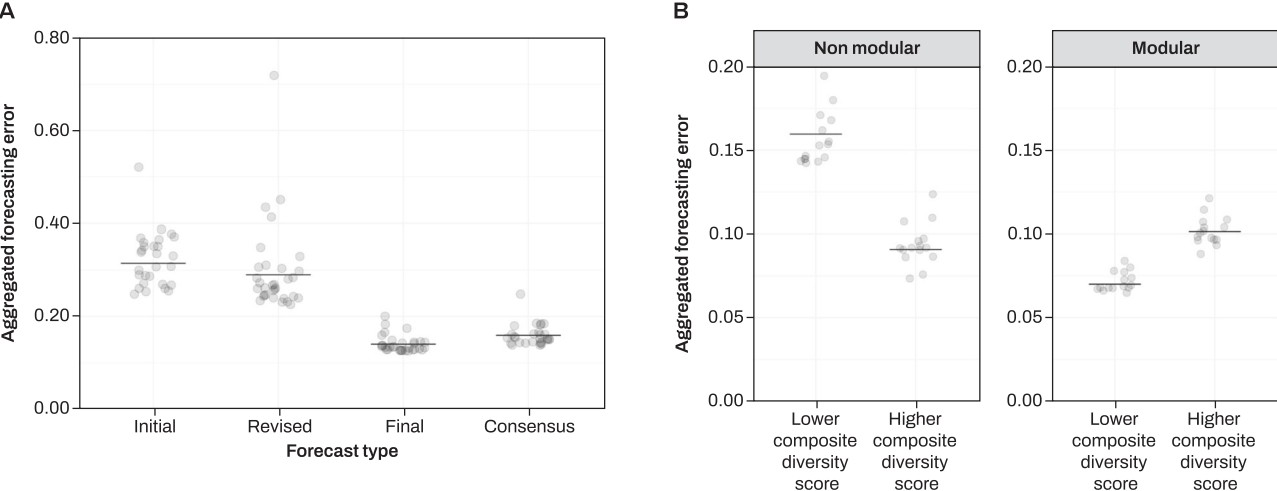

**Fig. 3 Group-level analysis.** Individual forecasts were aggregated for each forecast type, first within each group and then across groups in each treatment. **A** Partial residuals plot showing the effect of forecast type on aggregated forecasting error (measured in Brier scores). Lower numbers represent higher accuracy. Solid lines represent model fit. **B** Partial residuals plot showing the effect of diversity and modularity on aggregated forecasting error. Solid lines represent model fit. Notice that the graphs have been plotted onto the original error scale. See Supplementary Fig. 11 when using a logit link. Source data are provided as a Source data file.

similar characteristics is thought to create insulated online information bubbles. There is growing concern that this tendency might have negative long-term consequences on political and democratic institutions, as citizens form partial or inaccurate representations of the world. Although we cannot answer these important questions with our study, we tried to characterize the effect that interacting with peers who differ along an arbitrary large profiling space has on the forecasting accuracy achieved by in-expectation-identical people (core segment participants) as a function of group size. We provided preliminary evidence that the ability of an online collective to rapidly gather information to predict difficult geo-political events may be coupled with their digital ecosystem. People's shared traits did not predict a priori how correlated their beliefs about world events were. Instead, belief coupling happened only after they interacted with their unique information silos via their web browsers. Forecasts became correlated only after online browsing, and proportionally to people's similarity on our multi-trait profiling space. In other words, our operationalization of trait similarity had measurable effects on the online information a group could tap into. This is in contrast with offline settings, where trait diversity does not directly impact information diversity[23,25,26,29,37,38]. The use of an experimental methodology bypasses the limitations of observational approaches, strengthening causal inference[25,26,30,53]. Trait similarity in our experiment largely captured participants' variability along interpretable ethnic-cultural and socio-political variables (Supplementary Figs. 13–18). Arguably, these features affect political judgments and the type of content that a person is likely to retrieve online. Our findings raise worries that these features may be used by search engines to skew information retrieval during online searches, with measurable effects on collective performance. This effect was not among our preregistered hypotheses so we warn caution in overinterpreting this finding. Future studies should attempt a replication.

Our findings also suggest the importance of diversity in online settings characterized by large collectives. Given the difficulty and domain specificity of the questions in our experiment, increasing diversity may have increased the chance that at least one of the participants in a group could, for example, recall what a Loya Jirga is and make an informed guess. This effect would be more pronounced in a large group than a small group. To illustrate this,

imagine asking a group of scientists this question: "Is *Variola major* likely to become a more life-threatening virus than Coronavirus before 2030?". If we select a discipline at random, and then make large or small groups they would be unlikely to know what *Variola major* is and would guess Yes with some probability greater than zero. If we randomly choose scientists across disciplines to create groups, small groups do not do much better than groups from a single discipline because the likelihood of containing a virologist is small. However, the chances of finding a virologist increase with group size and a finite number of academic disciplines. If there happens to be a virologist, they can trivially identify the answer to this question as No (this virus causes small pox, a disease that the World Health Organization declared eradicated in 1980). Similarly for political questions, imagine we have a set of questions from across a large range of countries or cultures, all of which are obviously unlikely to anyone with domain knowledge. Diversity would improve forecasting in large, but not small groups, because large groups have an increased chance of containing an expert. Critically, because the base rate probability of the events is low (Supplementary Table 1), Brier error will be high in anyone without domain knowledge that assumes the events have closer to equal probability of occurring. Although this logic nicely explains the beneficial effect of diversity observed in large groups, it lacks explanatory power in other respects. First, it does not explain why we observed a symmetrical effect in small groups instead of no effect at all (Figs. 2B and 3B). Second, it does not explain why differences among groups largely emerged after the revision and social stages rather than during initial guesses. Finally, it is unclear why performance variability remained similar between large diverse and homogeneous groups, notwithstanding a supposedly different concentration of domain experts (Fig. 4). Thus, although these statistical considerations are certainly relevant, technological (individuals interacting with their search engines), and social (individuals interacting with each other) aspects are also an important part of the story. Importantly, alternative measures of diversity and more theory-driven profiling should be considered in the future to address these concerns. For the scope of our paper, however, the specific implementation of composite group diversity was not as important as its functional value in influencing information foraging and error distributions in online groups. Characterizing measures

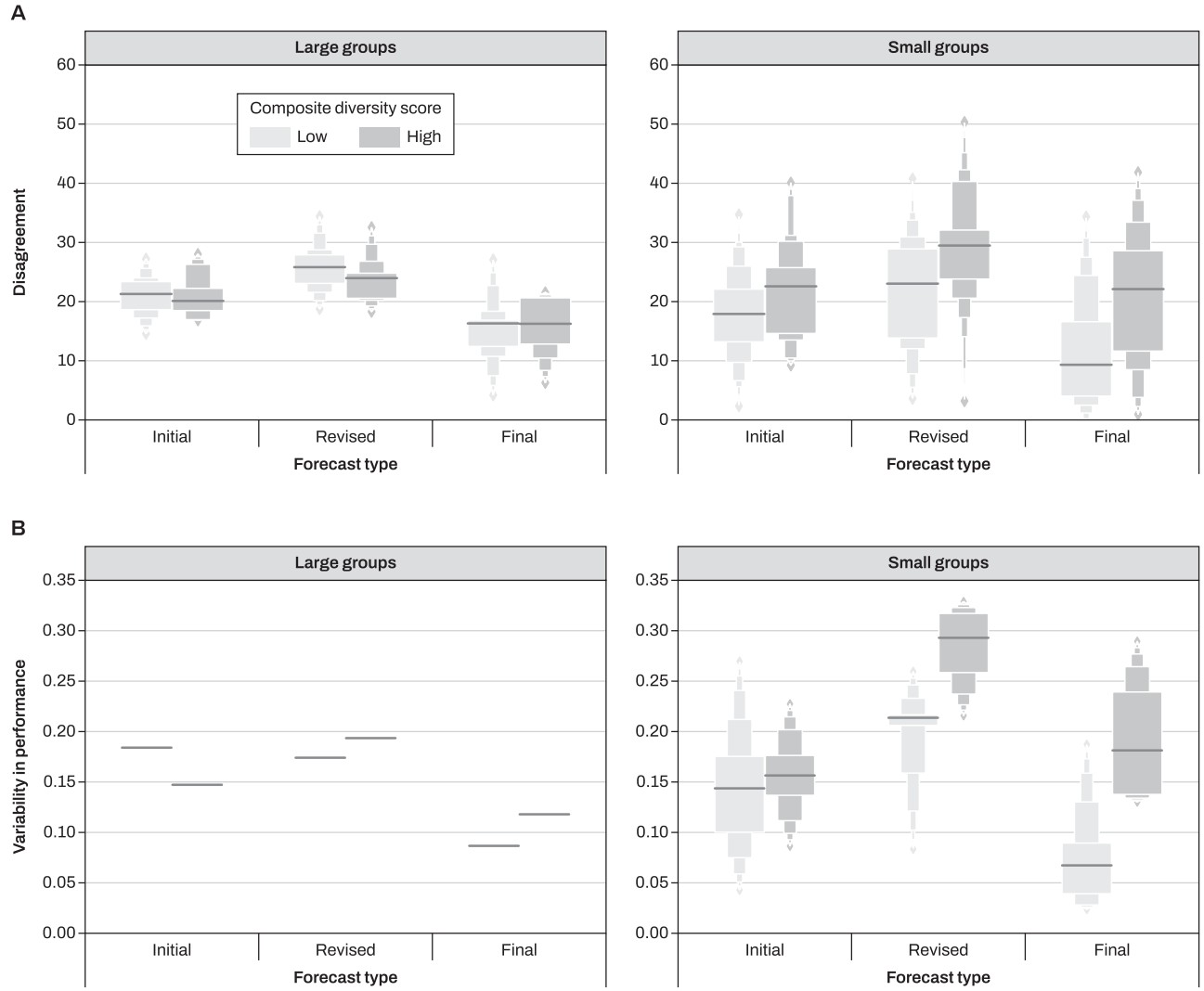

**Fig. 4 Disagreement and variability in performance. A** Distributions of opinion disagreement as a function of forecasting stage, group trait diversity and group size. Opinion disagreement is calculated as the standard deviation over group members' forecasts. **B** Performance variability as a function of forecasting stage, group trait diversity, and group size. Performance variability is the standard deviation over average individual performance in a group. Larger values indicate that a group contains members who are very good and members who are quite poor (on average across the eight IFPs). Notice that a single value of performance variability exists for large groups, but not for small groups ($m = 6$ and $m = 4$ for small low and high diversity groups, respectively). Notice also that for both panels consensus forecasts were removed because, by definition, they did not produce meaningful variation in these measures. Box areas correspond to distribution ideal tail areas of 0.50, 0.25, 0.125, 0.0625 (ref. [65]). Source data are provided as a Source data file.

of group diversity is a research field in its own right. We recognize that our method is not perfect and caution should be used when trying to generalize our results.

Investigating collective decisions under extreme conditions is highly informative. Many decisions faced by intelligence analysts, as well as normal people everyday are characterized by weak signal, uncertainty, time pressure, or short collective attention, namely all conditions under which rational deliberation is least effective[11,12,54]. The specific forecasting problems asked in the task were a random subsample of forecasting problems that were selected by a national forecasting tournament (Hybrid Forecasting Competition) to be a representative sample of professional geo-political forecasting. They required domain knowledge that participants were unlikely to possess prior to online browsing. This feature served a precise design purpose. The specificity of the forecasting problems ensured that group discussions were driven by the content that was collectively retrieved online rather than biased by what participants knew in advance. Group members had only a short amount of time to forage for relevant online

content. The ability of a group to collectively search relevant information in parallel was thus, arguably, more important than the ability of each individual to search any piece of information thoroughly. Finally, another thing to notice is that most events did not occur (Supplementary Table 1). This is not uncommon in forecasting. Rare events are often the most consequential and difficult to predict, as the covid-19 pandemic shows. Being able to predict rare events resides at the heart of accurate forecasting[49,55]. In these circumstances, an unspecific bias toward deeming events unlikely to occur would generally pay off (in the short term), and generate few highly consequential mistakes. To rule out the confound of an unspecific bias, we ran a signal detection analysis that indicated that people did not show any bias toward uncritically deeming events as rare during their initial forecasts (Supplementary Fig. 9). In later forecast stages, it is unclear why an unspecific tendency toward answer low probability (confidently believing the events were unlikely) would emerge from online browsing or social interaction. Social interaction is known to extremize initially held individual opinions, a phenomenon

known in psychology as risky-shift[56]. Thus, if anything one would expect social interaction in our experiment to pull initial predictions toward 0 and 100% symmetrically. Instead, group discussions seemed to adjust initial predictions intentionally toward the correct response. Furthermore, the unspecific bias explanation does not account for the interaction between group diversity and group size observed. Manual labeling of chat conversations revealed that about half of people in each group had at least some knowledge about each topic, and conversations mainly revolved around evidence in favor or against each option. Although it is difficult to disentangle whether domain-specific knowledge was due to prior beliefs or online browsing, the former explanation is unlikely given that initial forecasts were distributed around chance level (Supplementary Fig. 9). We thus conclude that the observed accuracy improvement was more likely due to online browsing and group deliberation, rather than an unspecific bias toward reducing probability.

In line with recent work in collective behavior, we find that when decision makers are not independent (as in this task) group accuracy can benefit from a reduced group size and increased modularity[15,41,43,44,57]. Research in social learning[58] has shown that group outcomes are affected by a complex interplay among several factors, including learning strategies, task complexity, modularity, and network structure. The present study showed how two factors that independently reduce correlated errors, namely group composition and modularity, can interact in unexpected ways[14,15,20]. To characterize this novel interaction, we described information aggregation using a range of exploratory analyses, such as within-group disagreement (Fig. 4A), convergence speed to consensus forecast (Supplementary Material §6) and performance variability among group members (Fig. 4B). Among these variables, performance variability—often a prerequisite for good group performance in the literature on collective intelligence[51,52,59]—may help understand how our treatment influenced information aggregation in our task.

Notwithstanding the value of these results, we would like to raise a word of caution. In particular, as specified in our preregistration, we had no expectations on the direction of the interaction between group composition and group size before testing our model. Similarly, some analyses were exploratory in nature and cannot be used to draw definitive conclusions. Future studies will need to address whether the result can be replicated. If so, our results suggest that, given the difficulty in reducing the impact of homophily and self-assortativity on the Internet, decision makers may try instead to increase its modularity. Addressing the ethical considerations in this debate is beyond the scope of this paper, but an equally important avenue of investigation[60].

## Methods

**Procedure**. The study was approved by MIT Institutional Review Board. Participants ($N = 193$, Supplementary Tables 2 and 3) gave informed consent before joining the study. Three days before test (pretest), participants answered a battery of demographic, cognitive, and personality questions that was used to map them on a multidimensional space $\Theta$. We used an unsupervised clustering algorithm (DBSCAN) to label participants as belonging to the center mass of the distribution (core segment) or its tail (inner and outer segments, Fig. 1A). This structure was already visible on a low-dimensional projection of participants on the first two principal components of the data (Supplementary Fig. 18).

We manipulated group composite diversity (low vs. high) and crowd modularity (low vs. high; Fig. 1B). Core participants (~50% of our initial sample) were randomly assigned (a) to work with either close (inner segment,~25% of our sample) or distant (outer segment, ~25% of our sample) participants on the feature space, and (b) to work in small (~5 people) or large (~25 people) groups (Fig. 1C). During the experiment (test phase), participants answered eight IFPs, randomly selected from a larger pool of binary real geo-political forecasting problems released within IARPA's Hybrid Forecasting Competition and unresolved (i.e., whose solution was unknown) at the time of the experiment. The exact problems selected were not preregistered. For each IFP, participants went through three timed consecutive stages. During stage one, participants answered a binary forecasting problem (Supplementary Table 1) and had to enter an initial private forecast off the top of their heads (initial forecast). During stage two, they had to search relevant information online, using their browser, and enter a revised private forecast (revised forecast). Finally, during the third and last stage, participants discussed in real time their views using an inbuilt chat (Fig. 1C). During this stage, participants had to agree on a joint forecast (consensus forecast), as well as giving their final private forecast (final forecast). Notice that although consensus forecasts in a group had to be the same final forecasts could differ, thus allowing us to capture residual disagreement existing between group members after interaction had taken place. Participants were rewarded both for their time and—~6 months later (post-test)—when the ground truths were revealed—for accurate predictions. Performance was evaluated using Brier scores, a quadratic error score used in forecasting for its proper scoring properties, i.e., a scoring rule incentivizing honest responding. For a binary question, a Brier score is computed as:

$$b = (o - p)^2 + (\bar{o} - \bar{p})^2 \quad (1)$$

where $p$ represents the predicted event probability (range [0, 1]) and $o$ is the indicator variable for the observed event (0: the event happened; 1: the event did not happen). $\bar{p}$ and $\bar{o}$ represent complementary probabilities. A Brier score of 0 represent a fully predicted event (i.e., no uncertainty), while a Brier score of 2 represents a gross forecasting error (the forecaster predicted with absolute confidence the event would occur and it did not, or viceversa). Notice that Brier scores measure second-order accuracy, meaning that they punish over- (and under-)confidence rather than number of incorrect binary judgments. An improvement in Brier score represents a more precise probabilistic forecast, which might not necessarily reflect how often a participant is right (first-order accuracy). For these reasons, Brier scores represent the standard in forecasting[49,61,62].

**Analyses**. Errors were fitted with multilevel generalized linear mixed-effects models (GLMM) with Gaussian log link function. The results are robust across alternative link functions, like probit and logit (Supplementary Table 17, and Supplementary Figs. 10 and 11). All analyses, unless specified, were limited only to participants who fell in the core segment (i.e., test participants), as these were the only ones to whom the randomization procedure applied. This allows us to draw causal inferences on the effect of our manipulation, as all core participants were equal in expectation. Our main analyses corresponding to our preregistered hypotheses are reported in Table 1A and D. They included at the individual level the effect of forecast type, and the aggregate level the effect of composite diversity and size assignment. To provide a full picture, we complement the main analyses with the effect of the manipulation on individual errors (Table 1B) and the effect of forecast type on aggregate errors (Table 1C).

Also according to our preregistered hypotheses, we analyzed within-group disagreement at each stage of the experiment (Supplementary Table 15). Disagreement was defined as the standard deviation of the forecast within a group, broken down by forecast type and condition. We also run a set of exploratory analyses on chat data, aimed at understanding how individuals integrated private information to reach a consensus within their group (see Supplementary material §5-6).

**Statistics and reproducibility**. The experiment was repeated only once. A pilot experiment had been previously discarded (data never analyzed) due to a bug in the web application.

**Preregistration material**. Preregistration material is available via AsPredicted.org: https://aspredicted.org/9m6df.pdf.

**Reporting summary**. Further information on research design is available in the Nature Research Reporting Summary linked to this article.

## Data availability

Research data supporting the findings of this study have been deposited in Open Science Framework. N.P., A.R., and I.R. (July 6, 2020). Modularity and composite diversity affect the collective gathering of information online. Data can be retrieved using the permanent link: osf.io/wb538. A Reporting summary for this article is available as a Supplementary Information. Source data are provided with this paper.

## Code availability

Code to replicate analysis and figures supporting the findings of this study have been deposited in Open Science Framework. N.P., A.R., and I.R. (July 6, 2020). Modularity and composite diversity affect the collective gathering of information online. Data can be retrieved using the permanent link: osf.io/wb538.

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

## Acknowledgements

This research is based upon work supported in part by the Office of the Director of National Intelligence (ODNI), Intelligence Advanced Research Projects Activity (IARPA), via contract number 2017-17061500006. The views and conclusions contained herein are those of the authors and should not be interpreted as necessarily representing the official policies, either expressed or implied, of ODNI, IARPA, or the U.S. Government. The U.S. Government is authorized to reproduce and distribute reprints for governmental purposes notwithstanding any copyright annotation therein. The authors would like to thank Zhaozheng Alice Jin for her contribution to the development of the experimental web application; Dr. Nick Obradovich for his support throughout this work; and Dr. Hans van Dijk for sharing his data and insights.

## Author contributions

Conceptualization, data curation, investigation, methodology, and project administration: N.P.; formal analysis, software, validation, visualization, writing—original draft : N.P. and A.R.; funding acquisition: IR; resources and writing—review and editing: N.P., A.R. and I.R.; and supervision: A.R. and I.R.

## Funding

## Competing interests

The authors declare no competing interests.
