## [Peer Review File · Nature Communications]

Reviewers' comments:

Reviewer #1 (Remarks to the Author):

This Ms reports on a study that investigated the effects of group size and diversity on group performance. I find the results interesting but have several major questions regarding the underlying methods.

The Condorcet-theorem is only valid for very specific assumptions and has given rise to many misunderstandings and should best be cited in the context of a recent critical study (James Marshall et al. 2019 in eLife).

Also, I am not aware of Condorcet actually showing empirically that larger juries perform better in courts. I believe he only explained that it might be possible. But I admit that I haven't actually read him in the French original.

Page 1

"The learning strategies used to explore and exploit the solution space S interact with task complexity, or the "ruggedness" of S , to produce the wisdom or madness of interactive crowds (14, 15). Solitary learning performs well in simple tasks (e.g., characterized by single peaked S), but becomes increasingly more costly and unreliable in more complex solution spaces. "

I found this section hard to follow without reading the papers and specific figures in those papers that these statements refer to.

Lines 65-70

I take issue with the definition of task complexity. Why should complexity be higher if "information among judges is often correlated"? This correlation makes it less likely that you get a better result by integrating multiple judgements but it is not per se a measure of complexity – certainly none that I have ever heard of.

Again reference is made to the ruggedness of S – you need to explain this instead of merely throwing jargon around from previous papers.

I would question whether we need the term diversity in collective intelligence research at all. All we need to know is whether the errors are correlated (and to which degree). Why bring diversity (demography, cognitive level etc.) into it?

Lines 127-128

This diversity measure is a composite of many variables. It is obvious that an average Euclidian distance of such variables can be calculated. But does it mean anything? Did you test for robustness using a different measure than the average?

Lines 166-167

The measure of individual diversity is already rather vague but I didn't see any explanation of what the

measure for group diversity is. This worries me particularly in the context of changing group sizes. How do you compare diversity measures across different group sizes? This is far from trivial.

Lines 242-245

But the p-value is non-significant?!

Why try to explain something that is not significant – it is not a result anyway?!

Reviewer #2 (Remarks to the Author):

This is a very interesting and timely study. To my best knowledge, the main results are novel and relevant to various fields such as cognitive science, management, decision sciences, and social psychology, among others. While there are several take-home messages in the paper, I found most interesting the interaction between group size and diversity on prediction accuracy, the use of an experimental approach to study diversity, and the idea that forcing consensus leads to lower prediction error compared to mere social exposure.

I do have, however, one comment that might reveal a major issue (comment #1), which could potentially compromise the quality of the experimental procedure and therefore reduce the credibility of the results. I also have several other relatively minor comments and suggestions for secondary analysis. At the moment I remain unsure whether the issue in comment #1 could be addressed in a future revision, but the remaining ones should be easily addressable.

1- About the pre-registration. The abstract and methods indicate that this is a pre-registered study. However, after reading the document uploaded to AsPredicted.org, it seems that only very minor aspects of the hypotheses, experimental design, and analysis were actually pre-registered. The most worrying concern is that there is a major contradiction between what was hypothesized in the pre-registration and the hypothesis described in the Introduction. More specifically, before running the study the authors say they expected “diverse groups to perform better than homogeneous groups and small group-based aggregation to perform better than large group-based aggregation”. They also explicitly said that they did not have any “specific expectations on how the dimensions will interact with each other”. On the other hand, the current framing of the paper seems to suggest that the authors did have a clear hypothesis inspired in previous literature (line 119: “Based on theoretical background (...), we expect group diversity to interact with variables such as task complexity and group size”). So my question is: which one is it? Did the authors expect or did not expect the observed interaction between group size and diversity? If they didn’t expect it, as suggested by the pre-registration, then that part of the introduction should be edited, and all results should be toned down given the known risks of false positives in exploratory research. Moreover, the authors should clearly state that they did not find any evidence in the data supporting their two main hypotheses (a positive main effect of diversity and negative main effect of group size on forecasting accuracy, as described in the pre-registration, Table

3B). In addition, all analyses that do not appear in the pre-registration document should be labelled as exploratory (e.g., group-level analyses, linguistic analyses, etc.). Finally, many methodological details are simply absent in the pre-registration (e.g., the selected forecasting problems). All in all, as it is reported right now, the paper gives the impression of doing overly explicit HARKing, which is exactly the opposite of what one should expect from a pre-registered study.

2- About the Introduction. The authors cite and describe previous theoretical and empirical research that is completely unrelated to the current study. Not only this generates confusion about the scope of the paper but at certain moments it even gives the impression that they are bragging about their knowledge of seemingly complex theoretical formulations. For example, the current work has absolutely nothing to do with the exploration-exploitation trade off. The task used consisted in one-shot forecasting problems with no learning or necessity to balance exploration and exploitation strategies. The definition of a solution space S , where one looks for peaks in a landscape with different degrees of “ruggedness”, have little to do with predictions (e.g., see Hong & Page, 2008, for a mathematical formalization of prediction and cognitive tasks). The authors repeatedly mention how diversity and group size might interact with task complexity, something that hasn’t been systematically manipulated here (there’s only one level of complexity in the selected task). Finally, the authors claimed to have used “real-world questions” in order to avoid neglecting “external validity”. While the authors should be commended for using real forecasting problems, this speaks about the ecological validity of the experiment and not about the potential replicability outside the population where the experiment was performed (i.e., what is formally known as external validity).

3- About the selected forecasting problems. One possible concern with the selected problems is that 7/8 of them had the same correct answer (i.e., the event did not occur). While I understand that it is hard (or even impossible) to balance the correct answers a priori, one should be cautious about potential confounds in the data. For example, is it possible to disentangle accuracy from a bias to predict that events won’t happen? To analyse this possibility it would be interesting if the authors could show distributions of probability estimates and test for evidence of such a bias.

4- About the skewness of forecast distributions. The authors report the surprising result that searching for information online slightly reduced the prediction accuracy and hypothesize that this effect might be due to the skewness of the distribution of revised forecasts. One way to test this idea could be by using (instead of a t-test) a non-parametric test such as a Wilcoxon sign rank test for equal medians.

5- The authors came up with a heterogeneous definition of ‘diversity’ which lumps together 26 demographic variables plus aggregate scores of different cognitive and personality measures. One question that arises from the results reported in this study is whether all types of diversity equally contribute to the interaction between group size and diversity on accuracy. Putting it differently, is the kind of diversity that hurts small groups the same or different to the kind of diversity that improves the accuracy of large groups? How does the diversity in different demographic variables map onto the variance of opinions? I believe that answering those questions might help understanding the mechanism underlying the main result of this paper.

Reviewer #3 (Remarks to the Author):

This paper addresses the important topic of how group size influences collective intelligence in humans. While very well written, and interesting, there are some major issues that cause concern, especially regarding how the data are analysed and the statistics employed to do so.

Firstly the authors write in their abstract, and throughout the paper, that this study was pre-registered. This is, of course, commendable. However the registration appears to be very unspecific regarding the statistical tests to be employed (a vague list families of tests is suggested) and these are not associated in the pre-registration to any study variables. It would be beneficial to the reader to have an easier way to access the pre-registration information, and perhaps it should be discussed in the text. This ties in with a problem in the paper which is that more than 60 statistical tests are conducted making it difficult to determine whether this is indeed testing hypotheses generated before the study, or rather post-hoc interpretation following a huge number of applied tests. Alongside this, in places even non-significant results are described as though they are meaningful effects (most notably L241 P4 but also L402, p7). In some places it appears that two effects are described with a single set of test statistics (L324, L333, p6). Controlling for such a large number of tests in the statistics, and interpretation, is also largely lacking. This may actually impact how confident we can be in the central findings, which rely on significance values near $p=.01$ (L344-354).

Of most concern is that many of the papers key findings don't seem to hold up when simply considering first-order accuracy. The difference between the first-order and second-order analysis (Supplement 4 Table S10-S13) is fairly buried in the SI and the discrepancy only briefly discussed (L192 seems particularly important).

One possibility that could explain this discrepancy is that the continuous measures violate assumptions of variance with changes in group size and that this creates a biased estimate of central tendency. These issues need to be looked into before we can interpret the results. On a similar note, Brier scores are fitted with a log-normal GLM, but the data are inherently bound (between 0 and 2) and consequently this would be an inappropriate link function for the data type. This could, in principle, cause several important issues, including possibly artificially lowering p-values. The authors should consider this - why was a beta distribution, or a zero-one inflated beta, not chosen?

There are some presentation issues throughout. Important data are shown as bar graphs (I haven't seen dynamite plots for a while!) and this is not useful since it obscures important data. This is, by no means, always the case - and in many figures we do get a good representation of the data, but this should be made consistent throughout. Some figures lack Y-axes and/or units in general the variation in style and font is a bit jarring. It would be good to stick to a simple, clear and consistent presentation throughout (do we really need a comic-sans-type font for the figures?). Unfortunately some of the SI figures are

unreadable upon printing out and this should be corrected.

One issue of concern in the experiment is that the participants are given an extraordinarily short time which to search for appropriate information (90s). Why so short? Is there previous justification for this? It would intuitively seem way too short for the purposes of the experiment.

“Characterizing the dynamics in which information (and thus errors) is allowed...” - the problem here is the word “is” which is correct for “information”, but not for “errors”, which would be “are”. Best to restructure the sentence, something like “Characterizing the dynamics in which information is allowed (and consequently errors too), to flow between...”

Fig S2 seems to suggest brier scores can be negative with the density fits.

Effect sizes should be in interpretable units.

Figure S3 (Left, lo) is extrapolated well beyond the available data. It’s also not clear whether unequal sample sizes could drive the apparent difference.

The fits in Fig. S6 are fairly unconvincing and don’t look as though they’d reasonably match distributional assumptions.

Lo/Hi Low/High is confusing.

Overall this is an interesting work, but further consideration of the statistical analyses is required prior to publication.

Response to referees

Diversity promotes collective intelligence in large groups but harms small ones

Reviewer #1 (Remarks to the Author):

This Ms reports on a study that investigated the effects of group size and diversity on group performance. I find the results interesting but have several major questions regarding the underlying methods.

Reviewer's comment: The Condorcet-theorem is only valid for very specific assumptions and has given rise to many misunderstandings and should best be cited in the context of a recent critical study (James Marshall et al. 2019 in eLife).

Authors' response: This is true. We have made this clear in the text to avoid confusion in the reader, and cited Marshall's comprehensive work.

L38: "*Under specific assumptions, greater gains in accuracy can be expected when more independent judgments are pooled together* \citep{Armstrong2001, Marshall2019, Condorcet1785, Horowitz2002}."

Reviewer's comment: Also, I am not aware of Condorcet actually showing empirically that larger juries perform better in courts. I believe he only explained that it might be possible. But I admit that I haven't actually read him in the French original.

Authors' response: To the best of our knowledge, Condorcet offered only an analytical demonstration. Later studies have however validated his point:

- Saks, M. J. & Martie, M. W. (1997). A meta-analysis of the effects of jury size. *Law and Human Behavior*, 21, pp. 451-467.

- Horowitz, I. A. & Bordens, K. S. (2002). The effects of jury size, evidence complexity, and note taking on jury process and performance in a civil trial. *Journal of Applied Psychology*, 87, pp. 121-130.

We included Horowitz et al. paper in the manuscript, to correct the misunderstanding. (L40)

Reviewer's comment:

Page 1

"*The learning strategies used to explore and exploit the solution space S interact with task complexity, or the "ruggedness" of S , to produce the wisdom or madness of interactive crowds (14, 15). Solitary learning performs well in simple tasks (e.g., characterized by single peaked S), but becomes increasingly more costly and unreliable in more complex solution spaces.*" I found this section hard to follow without reading the papers and specific figures in those papers that these statements refer to.

Authors' response: We thank the review for their feedback. We agree that the section was dense with concepts. We have simplified the text accordingly and removed any mention to landscape problems:

L21: “Social information becomes particularly important under conditions of uncertainty and when the cost of making individual errors increases. Anecdotally, when uncertain we tend to look for advice, copy others or discuss with our peers. In a range of complex problems, social strategies achieves better performance than individual strategies, and increased use of social information has been observed in both human and non-human animals \cite{Toyokawa2019,Kendal2004, Morgan2012,Barkoczi2016, Wisdom2013, Rendell2010}.”

Reviewer’s comment:

Lines 65-70

I take issue with the definition of task complexity. Why should complexity be higher if “information among judges is often correlated”? This correlation makes it less likely that you get a better result by integrating multiple judgements but it is not per se a measure of complexity – certainly none that I have ever heard of. Again reference is made to the ruggedness of S – you need to explain this instead of merely throwing jargon around from previous papers.

Authors’ response: Similarly to the comment above, we agree with the reviewer that our previous definition was too obscure, as it relies on knowledge of the Kao et al 2014 paper. We have simplified the language accordingly:

L65: “Here, we are interested in forecasting geo-political events, whose ground truth is unknown at the time of the decision. Forecasting is notoriously difficult because it requires judges to monitor multiple interrelated cues (eg economic indicators, political news, local information etc.), and information is often correlated across judges (eg individuals might have access to the same news sources) \citep{Taleb2008}.”

Reviewer’s comment: I would question whether we need the term diversity in collective intelligence research at all. All we need to know is whether the errors are correlated (and to which degree). Why bring diversity (demography, cognitive level etc.) into it?

Authors’ response: We are grateful to the reviewer for raising this high-level point, which is good for discussion. It is true that on a purely prescriptive level, talking about diversity is not necessary (as correctly pointed out by the reviewer). However, our discussion in terms of diversity must be understood within the framing of our paper, which is interested in interactions in online settings. The current debate around this topic often mentions the problem of informational echo chambers and homophily in digital environment. We thus believe that the explicit use of the term is appropriate to foster discussion around this important topic.

Reviewer’s comment:

Lines 127-128

This diversity measure is a composite of many variables. It is obvious that an average Euclidean distance of such variables can be calculated. But does it mean anything? Did you test for robustness using a different measure than the average?

Authors’ response: We are grateful to the reviewer for this important observation, which together with similar comments by reviewer #2, made us conduct a series of new analyses (reported below) that try to better characterize our diversity measure. We believe it led to a better understanding of our diversity measure and it improved the overall manuscript:

- We ran a Principal Component Analysis on our pre-test questionnaire data, which has 29 pre-screening dimensions (named Theta in the manuscript). PCA has the benefit to decompose the variability of our diversity measure into few principal components that retain most of the information while reducing irrelevant or redundant information. We show that (a) There are strong components explaining most of the variance, which suggest the presence of structure in the data; (b) The first five components explain a large portion of the variance (~90%), we can thus safely focus on these without the risk of missing important patterns, as it is likely they might be driving our observed effects. (c) Importantly, each one of these five principal components captures meaningful different aspects of trait diversity, (ie. the pre-test variability in our population). E.g., the first PC seems to capture race, the second Eastern-Western culture, the third politics, the fourth sexual orientation, etc.

We report this analysis in the Supplementary Information (Figure S15-17) and refer to it in the main text:

L512: *“To better characterize our diversity measure, we decomposed it into principal components and found that five components---roughly capturing participants variability along race, east-western culture, political and sexual orientation---explained almost 90% of the total variance, suggesting these factors might be driving our observed effects.”*

- We report the distribution of pre-test questionnaire responses broken down by participant segment (core/inner/outer), Figures S18-20. This plot shows along which dimensions Core participants differed from Inner and Outer participants in their pre-screen responses. It shows how some dimensions (like years of education and race) showed variation along the different population segments, while others (like age) did not.
- We provide the code to reproduce an exploratory analysis aiming at understanding how variability along each pre-screening question (Theta dimensions) correlates with variability in the forecasts made, broken down by group (*diversity_manipulation_check.ipynb*). For the sake of space we do not report all 29 figures in SI. However, the same analysis can be performed on PCA transformed questionnaire responses, corresponding to the questionnaire projections on the first five principal components. In the SI, we plot within-group participants variation along the identified principal components against a group’s median initial opinion diversity (Figures S21) and consequently consensus forecasting error (Figure S22). Although no single principal component seems to explain the results reported in the main text, a few insights emerge: (a) homogeneous small groups tend to show lower opinion diversity than diverse small groups (b) a wider spread of forecasting errors is observed in small groups than large groups, suggesting greater noise (c) while the large diverse group shows reduced error than large homogeneous group, the relation is less straightforward in small groups.
- Finally, we attempt to test whether our continuous measure of diversity (within-group mean Euclidean distance) correlates with others suggested in literature. Many measures exist to compute diversity so we tried to compute the main measures reported by (Biemann et al.). Unfortunately it’s not possible to directly compare most of these measures because they normally apply to samples of single attributes (e.g. Gini

coefficient for incomes). Instead we are interested in multidimensional diversity. This can be calculated for standard deviation and coefficient of variation. We looked at both these measures for the people in *high* and *low* diversity groups and compared. The results confirm a larger variability in traits in our diverse groups.

STD in *low* diversity group = 0.8904

STD in *high* diversity group = 1.115

CV in *low* diversity group = -28.4113

CV in *high* diversity group = 30.1471

Breaking down the measure by group showed that STD was also highly and positively correlated with our original diversity measure ($r=.92$, $p<.001$). We report these figures in SI and main text.

Furthermore, we conclude by saying that exploring different ways of measuring diversity is a research field in its own right. While we tried our best, we recognize that this is a difficult task and open problem, and so should be considered when interpreting our results. We stress the importance of this point in the manuscript too:

L522: *“Our measure was highly correlated with standard deviation ($r=.92$, $p<.001$), an alternative measure also used on multi-dimensional data. Comparison with all alternative measures used in the literature however was difficult given that many of them use single attributes (eg Gini coefficient). Exploring different ways of measuring diversity is a research field in its own right, and while we tried our best, we recognize that this is a difficult task and open problem. Thus caution should be used when interpreting our results.”*

Reviewer’s comment:

Lines 166-167

The measure of individual diversity is already rather vague but I didn’t see any explanation of what the measure for group diversity is. This worries me particularly in the context of changing group sizes. How do you compare diversity measures across different group sizes? This is far from trivial.

Authors’ response: We appreciate the reviewer’s comment, and we will try our best to respond to this point. However, we must admit that we are unsure if we understood the reviewer’s intended meaning of the term ‘individual diversity’, so we would be grateful if the reviewer could correct us if we misunderstood. Diversity is by definition a group construct, as an individual by themselves cannot be ‘diverse’. Group diversity in our manuscript is operationalized by assigning group members to conditions depending on their reciprocal distance in the manifold Theta, where Theta is simply the vector of answers to our pre-screening questions. The labelling of individuals as Core, Inner and Outer via DBSCAN, represents where in the distribution of answers given by participants in the pre-screening questionnaire, the individual’s responses lie. With these labels we proceeded to create diverse groups (Core+Outer) and

homogeneous groups (Core+Inner). We then tested (1) the effect that working in a diverse group had on individual performance (Table 2A-B) and (2) the effect that using forecasts from diverse groups had on aggregated performance (Table 2C-D), where aggregated performance was simply the performance of the forecast obtained by median aggregating forecasts in each experimental condition.

To make sure that our group assignment procedure did indeed have the effects that we intended, we validated our diversity manipulation in two ways. First, by checking that indeed it produced groups whose mean euclidean distance (MED) along the space Theta differed (Figure S1). And second, that the interaction between diversity and group size reported in the main text could be replicated using such MED measure (Figure S3).

After a reviewer's suggestion, we now also have introduced a comparison with an alternative measure of diversity used in the literature, namely standard variation (along Theta). We find that MED and St.dev. are highly correlated ($r=.92$).

Finally, we agree with the reviewer that comparing MED across group sizes raises the question of how it affects the dependent variables of interest (see Biemann & Kearney 2010 for a thorough investigation on this matter). For this reason, we replicated the analysis in Supp. Info. (Table S5) with the group size correction suggested by Biemann & Kearney (2010). The results are in line with what reported in the main text: a positive effect on Brier scores of forecast type_final ($p=.007$), a negative effect of corrected MED ($p=.05$) and modularity ($p=.02$) and, importantly, an interaction between corrected MED and modularity ($p=.01$).

Reviewer's comment:

Lines 242-245

But the p-value is non-significant?! Why try to explain something that is not significant – it is not a result anyway?!

Authors' response: The reviewer rightly points out that this is unusual. In fact, we did not discuss this result in an early version of the manuscript. However, we included the explanation when we learned that some readers were confused that figure 2 (top panel) shows revised forecasts are higher than initial forecasts, and they kindly suggested to briefly comment on why. However, after two reviewers (#1 and #3) were puzzled by our dwelling over a null result, we decided to revert back to the original and do not comment on the result to avoid confusion. We appreciate the reviewer pointing this out, and are reassured by the consistency among the reviewers on this point.

Reviewer #2 (Remarks to the Author):

This is a very interesting and timely study. To my best knowledge, the main results are novel and relevant to various fields such as cognitive science, management, decision sciences, and social psychology, among others. While there are several take-home messages in the paper, I found most interesting the interaction between group size and diversity on prediction accuracy, the use of an experimental approach to study diversity, and the idea that forcing consensus leads to lower prediction error compared to mere social exposure.

I do have, however, one comment that might reveal a major issue (comment #1), which could potentially compromise the quality of the experimental procedure and therefore reduce the credibility of the results. I also have several other relatively minor comments and suggestions for secondary analysis. At the moment I remain unsure whether the issue in comment #1 could be addressed in a future revision, but the remaining ones should be easily addressable.

Reviewer's comment:1- About the pre-registration. The abstract and methods indicate that this is a pre-registered study. However, after reading the document uploaded to AsPredicted.org, it seems that only very minor aspects of the hypotheses, experimental design, and analysis were actually pre-registered. The most worrying concern is that there is a major contradiction between what was hypothesized in the pre-registration and the hypothesis described in the Introduction. More specifically, before running the study the authors say they expected "*diverse groups to perform better than homogeneous groups and small group-based aggregation to perform better than large group-based aggregation*". They also explicitly said that they did not have any "*specific expectations on how the dimensions will interact with each other*". On the other hand, the current framing of the paper seems to suggest that the authors did have a clear hypothesis inspired in previous literature (line 119: "*Based on theoretical background (...), we expect group diversity to interact with variables such as task complexity and group size*"). So my question is: which one is it? Did the authors expect or did not expect the observed interaction between group size and diversity? If they didn't expect it, as suggested by the pre-registration, then that part of the introduction should be edited, and all results should be toned down given the known risks of false positives in exploratory research.

Authors' response: This is a very important point that the reviewer raises. Indeed, at the time of the pre-registration we did not have any specific prediction on the *direction* of the interaction between the two manipulated variables (diversity and modularity), although we suspected that they might interact. The mentioned literature allowed us to recognize these two variables as important factors for group outcomes and information correlation across judges. However, we did not anticipate that small groups would have been negatively impacted by diversity more than larger groups. We have rephrased the introduction and reiterate this concept several times accordingly, as the original phrasing was confusing for the reader:

L121: "Based on this theoretical background and the evidence reviewed above, we expected that group diversity and group size should contribute to group outcomes in complex tasks (no predictions were made regarding the direction of a possible interaction)."

AND

L229: "*We predicted an effect of diversity and modularity on aggregated forecasts after social interaction, but were agnostic about the direction of any possible interaction.*"

AND

L371: "*Finally, we found an interaction between the two terms whose direction we did not predict*"

And so on.

Reviewer's comment: Moreover, the authors should clearly state that they did not find any evidence in the data supporting their two main hypotheses (a positive main effect of diversity and negative main effect of group size on forecasting accuracy, as described in the pre-registration, Table 3B).

Authors' response: Relative to this second concern, we would like to offer a counter-argument. As the pre-registration suggests: *"Following Navajas et al. methodology, we expect that aggregating consensus forecasts and final individual forecasts (see below) across small groups will lead to better forecasting accuracy than aggregating within groups or over members of large groups. We expect diverse groups to perform better than homogeneous groups and small group-based aggregation to perform better than large group-based aggregation."* [our emphasis]. In other words, we expected the effect of diversity and modularity to show only at the group-level, thus after aggregating individual forecasts (and following the aggregation procedure described in Navajas et. al.). Table 3B mentioned by the reviewer reflects only the improvement in accuracy observed at the individual level. Observing an effect here would tell us that interacting with others within diverse and/or large groups is beneficial for the individual within that group. This effect, although interesting, was not predicted in our pre-registration (and now clearly explained in the manuscript, e.g., L282). The prediction mentioned in the pre-registration refers more appropriately to Table 3D, which shows that indeed, aggregating forecasts across members of diverse and modular groups is beneficial for forecasting accuracy. To avoid confusion, we changed the phrasing in the text to mark each analysis as exploratory or predicted, accordingly.

Reviewer's comment: In addition, all analyses that do not appear in the pre-registration document should be labelled as exploratory (e.g., group-level analyses, linguistic analyses, etc.). Finally, many methodological details are simply absent in the pre-registration (e.g., the selected forecasting problems). All in all, as it is reported right now, the paper gives the impression of doing overly explicit HARKing, which is exactly the opposite of what one should expect from a pre-registered study.

Authors' response: We thank the reviewer for raising this important issue, as it gives us fresh feedback upon which to improve the tone of the paper. We agree with them that the pre-registration is lacking some important details (e.g which forecasting problems were going to be used). Similarly, we clearly state that *"We have not agreed on how to analyze verbal communication among participants"*. For these reasons, we now clearly state which analysis are exploratory and which ones instead reflect a pre-registered prediction.

For example:

L241: *"To shed light on the underlying mechanisms, We analyzed within-group opinion conflict at each stage of the experiment. We predicted a reduction of conflict after online browsing and social interaction. We run a second set of exploratory analyses on chat data, aimed at understanding how individuals integrated private information to reach a consensus within their group."*

OR

L282: *"As an exploratory analysis, we then turned to analyze the effect of our experimental manipulation on individual forecasting accuracy (Table 2B-S4)."*

OR

L417: *“The difference in the effect of diversity on opinion conflict as a function of group size, in turn, affected the process of consensus reaching through online deliberation, as shown by a last exploratory analysis (see SI for full Methods).”*

Reviewer’s comment: 2- About the Introduction. The authors cite and describe previous theoretical and empirical research that is completely unrelated to the current study. Not only this generates confusion about the scope of the paper but at certain moments it even gives the impression that they are bragging about their knowledge of seemingly complex theoretical formulations. For example, the current work has absolutely nothing to do with the exploration-exploitation trade off. The task used consisted in one-shot forecasting problems with no learning or necessity to balance exploration and exploitation strategies. The definition of a solution space S , where one looks for peaks in a landscape with different degrees of “ruggedness”, have little to do with predictions (e.g., see Hong & Page, 2008, for a mathematical formalization of prediction and cognitive tasks).

Authors’ response: We are grateful to the reviewer for making us notice the previous manuscript had the ‘wrong’ tone. We think we have greatly lightened the reading of the text. We edited the introduction accordingly, so to remove mentions to landscape ruggedness and exploration/exploitation tradeoffs, e.g.:

L21: *“Social information becomes particularly important under conditions of uncertainty and when the cost of making individual errors increases. Anecdotally, when uncertain we tend to look for advice, copy others or discuss with our peers. In a range of complex problems, social strategies achieves better performance than individual strategies, and increased use of social information has been observed in both human and non-human animals”*

OR

L30 *“Characterizing the dynamics in which information is allowed (and consequently errors too), to flow between individuals can improve our understanding of group outcomes, and their resilience to sub-optimal solutions.”*

Reviewer’s comment: The authors repeatedly mention how diversity and group size might interact with task complexity, something that hasn’t been systematically manipulated here (there’s only one level of complexity in the selected task).

Authors’ response: After noticing this mistake, we removed all mentions to a possible interaction between our manipulated variables (modularity and diversity) and task complexity. It should now be clearer to the reader that we adopted one level only of (relatively high) task complexity

Reviewer’s comment: Finally, the authors claimed to have used “*real-world questions*” in order to avoid neglecting “*external validity*”. While the authors should be commended for using real forecasting problems, this speaks about the ecological validity of the experiment and not about the potential replicability outside the population where the experiment was performed (i.e., what is formally known as external validity).

Authors' response: We had not noticed this mistake in terminology and we have corrected every mention to "external validity".

Reviewer's comment: 3- About the selected forecasting problems. One possible concern with the selected problems is that 7/8 of them had the same correct answer (i.e., the event did not occur). While I understand that it is hard (or even impossible) to balance the correct answers a priori, one should be cautious about potential confounds in the data. For example, is it possible to disentangle accuracy from a bias to predict that events won't happen? To analyse this possibility it would be interesting if the authors could show distributions of probability estimates and test for evidence of such a bias.

Authors' response: This is a fair question to ask, and we have wondered ourselves how the distribution of problems' outcomes (due to unpredictable chance) might have affected our results. We concluded that there is no reason why the participants could be aware of such outcomes. However, we want to seriously address the reviewer's comment and report the distributions of forecasts for each of the 8 IFPs and for each participant in Supplementary Information (S11 & S12). We also provide the code to reproduce the distributions, given that the figure might be unreadable on print due to the large number of participants involved. We show, both at the participant and at the IFP level, that none of the distributions shows a bias toward always predicting a 0% outcome likelihood (the event will not occur). Regarding a specific statistical test, we are open to any suggestions.

Reviewer's comment: 4- About the skewness of forecast distributions. The authors report the surprising result that searching for information online slightly reduced the prediction accuracy and hypothesize that this effect might be due to the skewness of the distribution of revised forecasts. One way to test this idea could be by using (instead of a t-test) a non-parametric test such as a Wilcoxon sign rank test for equal medians.

Authors' response: We thank the reviewer for suggesting this analysis. We have run the Wilcoxon test between initial and revised forecasts and obtained a wilcoxon-test ($W = 99666$, $p\text{-value} = 0.00094$, unpaired), suggesting that indeed a differences in ranks exists (although the W . test, contrary to a GLMM, does not take into account the non-independence of individual data points obtained from the same individuals, and thus caution must be observed). Two independent reviewers (Review #1 and #3) however suggested removing that section altogether, given that it dwells too long on a non-significant result. We thus have opted for keeping the analysis only in the R Notebook publicly available on OSF.

Reviewer's comment: 5- The authors came up with a heterogeneous definition of 'diversity' which lumps together 26 demographic variables plus aggregate scores of different cognitive and personality measures. One question that arises from the results reported in this study is whether all types of diversity equally contribute to the interaction between group size and diversity on accuracy. Putting it differently, is the kind of diversity that hurts small groups the same or different to the kind of diversity that improves the accuracy of large groups? How does the diversity in different demographic variables map onto the variance of opinions? I believe that answering those questions might help understanding the mechanism underlying the main result

of this paper.

Authors' response: We thank the reviewer for their important observations, which together with similar comments by reviewer #1, made us conduct a series of new analyses (reported below) that try to better characterize our diversity measure. We believe it led to a better understanding of our diversity measure and it improved the overall manuscript:

- We ran a Principal Component Analysis on our pre-test questionnaire data, which has 29 pre-screening dimensions (named Theta in the manuscript). PCA has the benefit to decompose the variability of our diversity measure into few principal components that retain most of the information while reducing irrelevant or redundant information. We show that (a) There are strong components explaining most of the variance, which suggest the presence of structure in the data; (b) The first five components explain a large portion of the variance (~90%), we can thus safely focus on these without the risk of missing important patterns, as it is likely they might be driving our observed effects. (c) Importantly, each one of these five principal components captures meaningful different aspects of trait diversity, (ie. the pre-test variability in our population). E.g., the first PC seems to capture race, the second Eastern-Western culture, the third politics, the fourth sexual orientation, etc.

We report this analysis in the Supplementary Information and refer to it in the main text.

L512: *“To better characterize our diversity measure, we decomposed it into principal components and found that five components---roughly capturing participants variability along race, east-western culture, political and sexual orientation---explained almost 90% of the total variance, suggesting these factors might be driving our observed effects.”*

- We report the distribution of pre-test questionnaire responses broken down by participant segment (core/inner/outer), Figures S18-20. This plot shows along which dimensions Core participants differed from Inner and Outer participants in their pre-screen responses. It shows how some dimensions (like years of education and race) showed variation along the different population segments, while others (like age) did not.
- We provide the code to reproduce an exploratory analysis aiming at understanding how variability along each pre-screening question (Theta dimensions) correlates with variability in the forecasts made, broken down by group (*diversity_manipulation_check.ipynb*). For the sake of space we do not report all 29 figures in SI. However, the same analysis can be performed on PCA transformed questionnaire responses, corresponding to the questionnaire projections on the first five principal components. In the SI, we plot within-group participants variation along the identified principal components against a group's median initial opinion diversity (Figures S21) and consequently consensus forecasting error (Figure S22). Although no single principal component seems to explain the results reported in the main text, a few insights emerge: (a) homogeneous small groups tend to show lower opinion diversity than diverse large groups (b) a wider spread of forecasting errors is observed in small groups than large groups, suggesting greater noise (c) while the large diverse group shows reduced error than large homogeneous group, the relation is less straightforward in small groups.

- Finally, we attempt to test whether our continuous measure of diversity correlates with others suggested in literature. Many measures exist to compute diversity so we tried to compute the main measures reported by (Biemann et al.). Unfortunately it's not possible to directly compare most of these measures because they normally apply to samples of single attributes (e.g. Gini coefficient for incomes). Instead we are interested in multidimensional diversity. This can be calculated for standard deviation and coefficient of variation. We looked at both these measures for the people in *high* and *low* diversity groups and compared. The results confirm a larger variability in traits in our diverse groups.

STD in *low* diversity group = 0.8904

STD in *high* diversity group = 1.115

CV in *low* diversity group = -28.4113

CV in *high* diversity group = 30.1471

Breaking down the measure by group showed that STD was also highly and positively correlated with our original diversity measure ($r=.92$, $p<.001$). We report these figures in SI and main text.

Furthermore, we conclude by saying that exploring different ways of measuring diversity is a research field in its own right. While we tried our best, we recognize that this is a difficult task and open problem, and so should be considered when interpreting our results. We stress the importance of this point in the manuscript too:

L522: *“Our measure was highly correlated with standard deviation ($r=.92$, $p<.001$), an alternative measure also used on multi-dimensional data. Comparison with all alternative measures used in the literature however was difficult given that many of them use single attributes (eg Gini coefficient). Exploring different ways of measuring diversity is a research field in its own right, and while we tried our best, we recognize that this is a difficult task and open problem. Thus caution should be used when interpreting our results.”*

Reviewer #3 (Remarks to the Author):

This paper addresses the important topic of how group size influences collective intelligence in humans. While very well written, and interesting, there are some major issues that cause concern, especially regarding how the data are analysed and the statistics employed to do so.

Reviewer’s comment: Firstly the authors write in their abstract, and throughout the paper, that this study was pre-registered. This is, of course, commendable. However the registration appears to be very unspecific regarding the statistical tests to be employed (a vague list families

of tests is suggested) and these are not associated in the pre-registration to any study variables. It would be beneficial to the reader to have an easier way to access the pre-registration information, and perhaps it should be discussed in the text.

Authors' response: We agree with the reviewer that the pre-registration material is lacking some information that might be useful for the reader to interpret the results reported in the manuscript. For this reason, we now clarify in the main text, which analysis were part of our pre-registration (e.g. mixed effect models on aggregated forecasts) and which are instead just exploratory (e.g. linguistic and consensus reaching analyses). With this information readily at hand, the readers will be better able to determine what weight to give to each result.

Reviewer's comment: This ties in with a problem in the paper which is that more than 60 statistical tests are conducted making it difficult to determine whether this is indeed testing hypotheses generated before the study, or rather post-hoc interpretation following a huge number of applied tests. Alongside this, in places even non-significant results are described as though they are meaningful effects (most notably L241 P4 but also L402, p7). In some places it appears that two effects are described with a single set of test statistics (L324, L333, p6). Controlling for such a large number of tests in the statistics, and interpretation, is also largely lacking. This may actually impact how confident we can be in the central findings, which rely on significance values near $p=.01$ (L344-354).

Authors' response: Similarly to the previous comment, we think the reviewer has raised an important concern. To clarify these issues in the text, we have

- removed the interpretation of null effects (L241)
- corrected what was just a typo (L402)
- changed the more concise form of reporting two effects with the more verbose but clearer version (L324, L333)
- In order to avoid positive results due to multiple comparisons, we stick with the suggested best practice of performing post-hoc direct comparisons between levels of a factor only after finding an effect for that factor in a multilinear regression model (e.g. Table 2).
- Finally, we are unsure about how the exact number of statistical tests the reviewer refers to (>60) is obtained, but we tried to address this concern by clearly reporting in the main text which statistical analyses were expected given our pre-registration and which were instead unexpected or simply done with exploratory purposes. In order to avoid any misunderstandings, we would like to clarify that not each reported p-value comes from a separate analysis. Instead most reported effects come from a few multi-linear mixed-effect regression models, which already account for their multiple comparisons. For example, all reported effects on performance (Table 2) come from only 4 models (A to D in the table). Furthermore, rather than being 4 independent regression models, Table 2A and C represents more specific versions of models in B and D respectively. This, paired with our pre-registration of the hypothesis, should reassure the reader regarding the important issue raised by the reviewer.

Reviewer's comment: Of most concern is that many of the papers key findings don't seem to hold up when simply considering first-order accuracy. The difference between the first-order and second-order analysis (Supplement 4 Table S10-S13) is fairly buried in the SI and the discrepancy only briefly discussed (L192 seems particularly important).

One possibility that could explain this discrepancy is that the continuous measures violate assumptions of variance with changes in group size and that this creates a biased estimate of central tendency. These issues need to be looked into before we can interpret the results. On a similar note, Brier scores are fitted with a log-normal GLM, but the data are inherently bound (between 0 and 2) and consequently this would be an inappropriate link function for the data type. This could, in principle, cause several important issues, including possibly artificially lowering p-values. The authors should consider this - why was a beta distribution, or a zero-one inflated beta, not chosen?

Authors' response: We are grateful to the reviewer for this useful comment as it allows us to clarify an important point. We agree with the reviewer that the same effects reported for Brier scores are replicated also for binarized accuracy only when looking at individual accuracy (S10-S11), but not when aggregating forecasts (S12-S13). However, we would like to argue that this difference is not a concern, but rather unsurprising. These are our motivations for using Brier scores rather than binary accuracy:

- We used Brier scores as this is the measure that has emerged as a golden standard in many forecasting tournaments, like the IARPA Hybrid Forecasting tournament, which this study was part of. Brier scores are the standard measure in forecasting problems (Tetlock 2006) because they represent a fine-grained measure of error when making judgments of probability.
- Information is lost when passing from a granular to a more coarse measure, so binarized accuracy represents a coarse version of the same error. As a clear example, two forecasters S1 and S2 make a forecast, regarding the likelihood of an event X, of F1=51% and F2=99% respectively. The event happens. Arguing that both forecasters are correct because they both predicted X to be more likely than non-X misses an important difference between the two.
- This is also clear in our data. Making a prediction at chance (50%-50%) would result in a BS of $(0.50-1)^2+(0.50-0)^2=0.50$. As it can be seen in our data (Figure 2-3) most data points are below .5, suggesting that binarizing these judgments would have treated most data points as equal and thus obscured the subtle variation in forecast accuracy that we are interested in here.
- The possibility that different group sizes might have differently impacted the variance of the outcome variable, is not supported by our conflict analysis, which shows that forecasts conflict (a direct measure of variance) was not affected by group size ($p>.05$).
- A final observation is that when looking at binarized accuracy in aggregate forecasts (Table S13), but not when looking at Brier scores (e.g., Table 2D in the main text), models return a singular fit, suggesting that after coarsening the brier measure there is not enough variability to draw meaningful conclusions about the predictors. When looking at the individual level results Table S10-11 (more variable and numerous), the model can be fit without problem and the results match the ones found with Brier scores.

Nevertheless, the reviewer's point is correct and highlights an important point that needs to be made clearer in the manuscript: our manipulation does not affect how often people 'get it right' (order-I accuracy), but rather how *precise* their predictions are, and consequently the predictions we can make from them (order-II accuracy). To make this subtle distinction clearer we have changed the text appropriately:

L212: "*Notice that Brier scores measure second-order accuracy, meaning that they punish over- (and under-)confidence rather than number of incorrect binary judgments. An improvement in Brier score represents a more \textit{precise} probabilistic forecast, which might not necessarily reflect how often a participant is right (first order accuracy). For these reasons, Brier scores represent the standard in forecasting. \cite{Tetlock2006, Tetlock2015, Fleming2014}.*

”

Finally and most importantly, the reviewer raises a key issue, namely that alternative link functions should be considered, due to the different support of the outcome variable ($\text{Brier} \in [0, 2]$) and the log link function ($x \in [0, +\infty]$). Our motivations to choose the log normal distribution was for sake of consistency with the literature, which shows how errors in estimation tasks tend to be log-normally distributed (Kao et al. 2018). Yet, we want to take the reviewers' concerns seriously as we believe it is important to test alternative distributions to show the robustness of the results. We took the liberty of using a Logit normal and Probit normal distribution rather than the Beta distribution suggested by the reviewer, because Beta regressions are not supported to the best of our knowledge in mixed-effects models packages. Using a simple beta regression (e.g. *betareg* in R) would thus violate the assumption of independence due to the correlations existing between data points within individuals, groups or questions). The new probit and logit link functions have the same support as the Beta distribution, namely the $[0, 1]$ range, and are implemented in most mixed-effects packages. Using these functions, we replicate the same results obtained with the log-normal. We provide the code to reproduce the new analysis with probit and logit, we added the figures and tables relative to the logit analysis in SI (Figure S13-14, Tables S16) and referenced them accordingly in the main text (L233).

Reviewer's comment: There are some presentation issues throughout. Important data are shown as bar graphs (I haven't seen dynamite plots for a while!) and this is not useful since it obscures important data. This is, by no means, always the case - and in many figures we do get a good representation of the data, but this should be made consistent throughout. Some figures lack Y-axes and/or units in general the variation in style and font is a bit jarring. It would be good to stick to a simple, clear and consistent presentation throughout (do we really need a comic-sans-type font for the figures?). Unfortunately some of the SI figures are unreadable upon printing out and this should be corrected.

Authors' response: We appreciate the reviewer's attention to details as it gives us the opportunity to improve the quality of our visual material and the overall clarity of the manuscript. For this reason we have taken the following steps:

- Dynamite plot: We changed the dynamite plot to include the full distribution of data points.
- Missing Y-axes labels: This is because the title says what the y-axis is! We think this is a more elegant way to convey the information that reduces the clutter on the figure. However, given that this is a matter of personal taste, we are happy to go with the reviewer's suggestions, if our "title" view is still considered unclear.
- Font type: We changed the font type from Comfortaa to Archivo, as we thought this might be more in line with the reviewer's sentiment
- Font size: We increased by 40% the font size of most large figures composed of multiple panels in SI. This should hopefully be of help when printing the SI on paper.

Reviewer's comment: One issue of concern in the experiment is that the participants are given an extraordinarily short time which to search for appropriate information (90s). Why so short? Is there previous justification for this? It would intuitively seem way too short for the purposes of the experiment.

Authors' response: We totally agree with the reviewer on this issue and would have loved to have allowed participants to browse information for a longer time. The reason behind this choice is that we wanted to be consistent with the original Navajas paper in the number of problems asked to the participants (here and there, 8 questions) and stay within the recommended one hour duration to avoid participants losing concentration or disengaging with the task. To fit them within this time-window, we calculated about 7 minutes per question. As perhaps guessed by our pre-registration material, we were particularly interested in the social forecasts, rather than the revision forecasts. This was in part due to the fact that a few minutes would have been more beneficial (in terms of forecasting accuracy as well as potential interest of the results) when spent interacting with others than reading an extra article privately online. Furthermore, it is unclear whether an extra minute or two would have been sufficient to privately gather enough evidence to make a fully informed decision on such difficult topics.

We think clarifying the issue is important, so we changed the text in the "Full Methods" section:
L71: "To be consistent with \cite{Navajas2017}'s study, timing for question was calculated so to present 8 forecasting problems within the duration of the experiment (one hour)."

And few lines later:

L75: "*Although short, the goal of stage 2 was to simply prime people with the first information they could find in their digital sphere, rather than allowing a critical appraisal of the topic or the reading of in-depth articles on the subject.*"

Reviewer's comment: "Characterizing the dynamics in which information (and thus errors) is allowed..." - the problem here is the word "is" which is correct for "information", but not for "errors", which would be "are". Best to restructure the sentence, something like "Characterizing the dynamics in which information is allowed (and consequently errors too), to flow between..."

Authors' response: We used the suggested phrasing as it more clearly reflects what we had in mind.

Reviewer's comment: Fig S2 seems to suggest brier scores can be negative with the density fits.

Authors' response: Yes, this is the trade-off existing between density plots and histograms. Although density plots are recommended (due to the fact that they don't present the issues typically related to bin-centers and bin-edges) they can be problematic in bounded conditions like ours. We have thus opted for showing both options superimposed. The reader will gain in clarity what is lost in tidiness.

Reviewer's comment: Effect sizes should be in interpretable units.

Authors' response: We thank the reviewer for spotting an important missing piece of information. Although the estimates reported in the tables are on log scale, the plots show the models estimates on the response scale (Brier score) which are more interpretable. Thanks to the reviewer's suggestion we have added an extra column in Table 2 that shows the effect (in expected Brier score) that can be expected with each predictor. Furthermore, all continuous predictors were standardized before being entered in the regression models, so their coefficients can be compared. Categorical predictors (e.g. Modularity vs. Diversity) were declared as factors and their coefficients are also comparable. We state this more clearly in the 'Full Methods' section (L105).

Reviewer's comment: Figure S3 (Left, lo) is extrapolated well beyond the available data. It's also not clear whether unequal sample sizes could drive the apparent difference.

Authors' response: We agree with the reviewer that the variability on the x axis is different in the two panels. This is because, by design, there were only two large groups but 10 smaller groups. However, the number of data points is actually comparable in the two panels because the number of participants was kept (again by design) the same, namely 52 in each modularity condition (see Table S2). Thus unequal sample sizes should not cause concern. However, if this is still cause of confusion we are happy to take down the figure (and relative analysis), given that it is not pivotal for the interpretation of the main results, but rather an opportunity to test whether our results hold when parametrizing diversity.

Reviewer's comment: The fits in Fig. S6 are fairly unconvincing and don't look as though they'd reasonably match distributional assumptions.

Authors' response: We are grateful to the reviewer for the honest feedback. We agree with the reviewer that it might distract from the central message of the manuscript. We thus have removed it from the Discussion and the SI.

Reviewer's comment: Lo/Hi Low/High is confusing.

Authors' response: We thank the reviewer for their honest feedback. We changed the labeling in the main text accordingly to avoid any confusion.

Reviewer's comment: Overall this is an interesting work, but further consideration of the statistical analyses is required prior to publication.

Authors' response: We highly value the feedback received (and the time taken to do such a careful and thorough review) and we believe we have taken concrete steps toward clarifying some of the misunderstandings and clearing doubts regarding our procedure and statistical methods. In particular, we addressed 39 reviewers' comments, added 5 new citations, ran 8 new mixed effect models, ran one principal components analysis, compared our diversity measure with an alternative used in the literature, added 12 new figures, improved readability of the text, and clarity in the information communication style via figures and tables, both in the main text and Supplementary information. We hope this effort has provided further support to our main results. We believe the manuscript is in a better shape now than it was before the reviewing process.

REVIEWER COMMENTS

Reviewer #1 (Remarks to the Author):

By far my biggest problem with the paper is the measure of diversity. Measuring all and everything about a person and then have an unsupervised algorithm cluster it made me wonder what can actually be learnt from this approach.

Lines 132-133. Average Euclidian distance along those axes to measure diversity can certainly be calculated. The question is whether it means anything. I was wondering what makes average Euclidian distance such an appropriate measure and how it compared to other measures?

In other words, I was wondering how robust your results are to the use of different diversity measures.

How does your diversity measure, for example, map onto traditional ones?

Did you compose your groups also according to other (more traditional) diversity measures to investigate whether your results are robust?

The title makes the results sound like dogma but this finding is just one realization under particular circumstances. Other articles have found that diversity can also benefit small groups (Kurvers et al. 2016 PNAS) or that diversity particularly benefits intermediate groups (Kao and Couzin 2014 PSRB) – it all really depends on how you define diversity and what the accompanying circumstances are.

In view of this I would rephrase: “Diversity can promote... and may harm...”

I often wonder whether diversity in a decision-making context is even a useful term. Error is correlated to a certain degree between individuals and that is the only type of diversity that is meaningful for group performance. Every other kind of diversity (e.g. homophily, demographic factors) is only a proxy for this degree of independence.

You acknowledge on line 75 that diversity is a complex construct but nevertheless then delve into its discussion anyway – without defining diversity. This is not helpful.

I also wondered about your results in the context of recent findings by Kurvers et al. PNAS. They found that uncorrelated error (i.e. diversity) only mattered if both participants were of similar ability. If one participant is considerably better than the other then there can be no benefit from a group decision. Do you have enough decisions from individuals to measure ability and to see whether this applied to your small-group decisions as well?

Lines 375-378 The major result described here is supposed to be found in Figure 3b. However, in Figure 3 I could find no label for (b) in the caption.

The same is true for figure 2, I believe.

I also had some issue with the interpretation of the data because I am not sure whether you can conclude from your data that there was greater “conflict” if the standard deviation of diversity was greater. It could also mean that participants with greater diversity had different ways to articulate themselves or that there is greater variance in social dominance to name just a few potential factors. I

know that many variables entered your diversity measure but this doesn't mean they were controlled for in the actual discussion.

Cherry-picking the one you like seems a little selective. At best I would speak of "conflict-potential" because you never empirically measured any form of conflict.

Reviewer #2 (Remarks to the Author):

The manuscript has been substantially improved. I would like to thank the authors for their dedication engaging with all my previous concerns. I believe that, in the current version, the results and framing of the paper are much stronger compared to the first submission. I only have two final -relatively minor- comments, which should be taken as suggestions to make the manuscript even more clear (comment #1) and to justify the selection of five components in their PCA (comment #2).

1) Regarding the framing of the paper: The authors mention in several parts that they adopt the "well-established framework of social learning" (line 28, abstract) and that they look at diversity through the lens of "social learning and network science" (line 117). Even in the abstract, they suggest that they import knowledge from cultural evolution and ecology, but it remains unclear exactly what specific knowledge from those fields are informative to the current study. I would like to suggest the authors better explain exactly what they mean by "the framework of social learning" given that there are many theories/models of how people learn in social settings.

2) I appreciate and liked very much the PC analysis performed by the authors where they selected five components explaining a large proportion of the variance (~90%). Given that this is a 29-dimensional space and the remaining 24 dimensions explain ~10% of the variance, however, one could still have more than five meaningful components. For example, if the sixth component explained 9% of the variance, then this would be approximately 3 times larger than what one would expect simply by chance (roughly $1/29=3\%$). To formally select a given number of components, the authors could perform a random permutation analysis and show that from the sixth (or seventh, or eight,...) component the remainder amount of variance is consistent with what is expected by chance. This would help justifying the selected number of components in their analysis.

Reviewer #3 (Remarks to the Author):

Following revisions, there are still substantive issues with this paper that need to be addressed. The claims made are suggested to be generic, but there are many aspects of this work that are problematic, being based on very specific aspects (as detailed below) which makes it not possible to generalise the

results in the way the authors claim. They need to substantially tone down their claims and directly address the specificities of this work, as outlined below.

The paper begins with a bold claim about the role of diversity and group size in promoting collective intelligence. Diversity is, inherently a broad term, and it's unclear what it means in the context of DBSCAN-sorted groups that are forced into three clusters. Are these natural clusters? If not, how might other latent and un-captured structure interact with group size? The use of PCA to uncover the major axes of variance is useful, but it remains unclear what diversity means in the context of this study, and less so how these results could be considered general.

In addition there are issues with the notion of collective intelligence in this work. It is understood that the questions were selected as part of a broader project, yet the questions asked surround fairly obscure geopolitical events that we find it unlikely could be driven by domain knowledge of MTurk workers. For example, workers would need to know what a Loya Jirga is, why they convene, and the likelihood of events that would prompt such a convention in Afghanistan. It is very difficult to believe the initial answers for these questions were based on prior knowledge for all but a very few select participants. With the **very** brief period to search for information, it's also difficult to imagine they would go far beyond simply knowing what the question was asking. Collective intelligence requires some (albeit noisy) domain knowledge. Even if MTurk participants (unbeknownst to me) have a very deep knowledge of obscure geopolitical events, it is important to note that is a very restricted definition of collective intelligence and the title would be better worded as "forecasting geopolitical events" than "collective intelligence".

We are left with a claim the DBSCAN-defined diversity interacts with group size to impact the ability of MTurk participants to forecast unlikely (most did not occur) and obscure geopolitical events. The title's overly bold claim is, therefore, most likely unwarranted.

Further, given the ratio of events in the questions that did not eventually occur, it would seem to be inherently impossible to understand whether this represents accuracy versus a simple previous bias to choosing "no". Any claims of collective intelligence could equally be replaced with "preferring to answer 'no'". It is understood that this arose from an inherent constraint of asking prediction questions about real events, but note that this challenge of experimental design does not ameliorate the limitations to the validity of the conclusions. This possibility needs to be addressed directly, as do its consequences.

Beyond this, it is concerning the extent to which this paper relied on loose pre-registration and abundant researcher degrees-of-freedom. It would appear to any skeptical reader that this paper may be a result of HARKing a noisy dataset. We lack simple things like plots of raw brier score (not partial residuals) for each condition, and other useful information to make informed decisions. A plausible alternative explanation is that the results are an artefact of bias and convergence across conditions and have very little to do with collective intelligence or diversity.

Re: pre-registration

These clarifications do not go far enough. The statistical tests in the pre-registration list a large number of common statistical test families and provide far too many researcher degrees of freedom. Even with the added caveats, I do not feel it is appropriate to claim the statistical tests were sufficiently pre-registered. In general, they should include which tests/variables/etc.. will be used to test which hypotheses and criteria that will be used for evaluating the results.

Re: stats

There are approximately 60 tests of significance (e.g. p-values) throughout the text including those that are non-significant.

False positive rates and overfit will increase with model complexity. Between this and the researcher degrees of freedom afforded by the pre-registration it remains difficult to disentangle the key findings from statistical artefacts.

Re: first order vs. second order accuracy

The use of alternate (e.g. logit) link functions is valuable, but note that log link functions are used to model error in numeric estimation tasks is fundamentally different from discrete choice (even with probability) tasks.

the clarification on Brier scores as a dependent variable of interest is appreciated.

Re: quality of plotting

Adding same-color and difficulty to distinguish dots on top of a dynamite plot doesn't really help with the display of information. Suggest removing the bar entirely and perhaps replacing with a box and jittered-points. This is particularly true as the measure can (and does) go below zero.

In general the plotting aesthetics are still problematic. Some have white backgrounds, some have grey. Some have a y-axis, some don't. Colormaps seem somewhat random, and often have color-blindness issues (e.g. red/green in same plot).

Dynamite plots persist in the SI

Those giant grids of tiny plots in the SI are still fairly confusing and largely unreadable.

Re: the specificity of a task

Consistency with previous methods and constraints do nothing to ameliorate the scientific concerns regarding the oddness of the behavioral task.

Re: Negative density fits

This can be dealt with using the “clip” option in seaborn (which seems to be the plotting library used)

REVIEWER COMMENTS & AUTHORS' RESPONSE

Reviewer #1 (Remarks to the Author):

By far my biggest problem with the paper is the measure of diversity. Measuring all and everything about a person and then have an unsupervised algorithm cluster it made me wonder what can actually be learnt from this approach.

Lines 132-133. Average Euclidean distance along those axes to measure diversity can certainly be calculated. The question is whether it means anything. I was wondering what makes average Euclidean distance such an appropriate measure and how it compared to other measures?

In other words, I was wondering how robust your results are to the use of different diversity measures. How does your diversity measure, for example, map onto traditional ones?

Did you compose your groups also according to other (more traditional) diversity measures to investigate whether your results are robust?

AUTHORS: We understand the reviewer's concern. We are happy to offer further proof of our reasoning and contribution. We hope the reviewer can find value in this approach as we do. We provide below a summary of our approach in addressing this multifaceted concern, and try to unpack these questions in more details.

We addressed the reviewer's concerns using several parallel approaches, which together can clarify our work. First, we provide a clearer motivation of why using this approach may be important and why some readers may find it valuable. Second, we stress the fact that we use Euclidean distance only as a tool (a proxy) to sort people into groups that differ in informational diversity, we don't claim that this measure is intrinsically meaningful or 'correct' measure of diversity. Third, we characterise the meaning of Euclidean distance in our study by looking at the relation between this metric and forecast correlation. Fourth, we characterize *post hoc* the multi-trait survey data this metric is based upon to understand variance in our population. This allows us to understand what dimensions of diversity the Euclidean distance metric relied upon. Fifth, we tone down the paper (starting from the title and abstract) to highlight that these results apply only to a specific task and context, and caution should thus be adopted in generalizing our results to other aspects of diversity. Sixth, we investigate the correlation between Euclidean distance and alternative measures of diversity when multiple features are present.

Measuring all and everything about a person and then have an unsupervised algorithm cluster it made me wonder what can actually be learnt from this approach.

AUTHORS: This is a very valid concern and we think that, with appropriate theoretical motivations and post-hoc analyses, we can learn a lot from this approach. Below is a detailed motivation for selecting this approach and an explanation of how the personal features were selected in the survey.

The motivation. The reasons for choosing a multi-dimensional approach to diversity was not casual (although perhaps uncommon to psychologists). It was rooted in our interest in understanding how algorithms commonly used in online spaces to segment populations and customize content, influence our collective ability to retrieve task-relevant information online, and cooperate with others to solve complex problems. We notice that most algorithms used nowadays segment users online based on big black boxes, i.e. unsupervised clustering algorithms taking in a multidimensional array of features and using it to customize people's experience (Joshi et al 2011, Mei et al 2008). This practice is popular because digital traces are easy to obtain and contain a cornucopia of information about people (Lazer et al 2009, *Science*). Controversial studies like Kosinski and colleagues' 2013 Facebook study show how easy it is to infer multiple personal traits from digital traces. Feature selection in these cases follows a pragmatic rather than theory-driven approach, often including demographics, social, behavioral and geolocational indicators, with the only criterion being the increase or decrease in click through rates. This "measure all" approach certainly seems to lack a deep understanding of the causal paths leading to the desired result. However, it is undeniable to the AI practitioner that this approach works in practice to produce significant behavioral shifts in products, advertisement and politics, as the Cambridge Analytica scandals and many empirical investigations have unveiled. Web search engine recommendations can shift voter preferences by 20% or more in indecisive individuals, with the effect scaling for certain demographic groups (Epstein 2015). The public conversation typically focuses on two major issues, the filter bubble generated by black box recommendation algorithms, and the opinion manipulation of voters' behavior. Here, we are interested in a less bespoke but equally important effect, namely our collective ability to retrieve and effectively use information for accurate decision-making.

If the above can be considered the practitioner's approach, a radically different approach is taken by psychologists, who have for a long time been interested in understanding the relationship between diversity and group performance with a focus on causal paths and consistent theoretical frameworks. This research program has provided a range of results and observations in a more localized, controlled approach. The literature is very vast and we certainly do not want to put a definitive answer to this very fundamental question (we indeed changed the title that sounded too strongly conclusive). Indeed, it is so vast that the range of results found often spans from results showing that diversity decreases performance to results showing that diversity increases group performance. As correctly pointed out by the reviewer, this is because "diversity" is an easy to understand umbrella term that is often attributed to different features of diversity considered. E.g., two common factors invoked to explain this apparent contradiction is the difference between demographic and informational diversity, with the general understanding that the first one is "bad" while the second is "good" (deOliveira 2018, vanDijk et al 2012, Horwitz et al. 2007). Importantly, the approach usually followed by psychologists is to investigate individual axes of diversity, e.g. racial (Antonio et al 2004), age and gender (Wegge et al 2008). This often allows psychologists to carefully characterize the causal mechanisms underlying the observed effects, even though the debate is far from over (van Dijk et al 2012).

Our approach tries to integrate the two. We want to use the computational social science perspective to bridge intuitions and practices across these two research programs, to 1. Understand whether we can apply carefully randomized protocols to “data-science-inspired” routines; and 2. Collect experimental evidence that general purpose clustering algorithms may skew our ability to solve complex tasks together, when applied to online group decision making contexts. We believe the question is timely and a conversation among scholars and the public is needed.

We notice that in a digital space where behavioral, psychological and demographic traits are transformed into differential access to information (e.g., “The filter bubble”), this tension between demographic and informational diversity, studied by psychologists, may not apply or may need to be revisited. The news is full of stories of people noticing patterns in their digital applications that reflect their physical appearance¹. We can expect that these are only the tip of the iceberg of a set of deeper patterns that are ongoing but that people may fail to notice, which are caused by the intrinsic multi-dimensionality of the profiling space considered by these algorithms.

Notwithstanding the (very important) concerns about the risk of solidifying biases and assumptions about minority groups, these same user profiling techniques may equally create the risk of skewing information sources, like the news, that inform people’s judgements. This can consequently affect their ability to solve complex problems, like forecasting problems and decision-making, especially when these problems refer to relatively obscure events like in our experiment.

Please notice that in our experiment, questions were selected so that people did not have much prior knowledge (another reviewer asked why this was the case) because we did not want prior biases, strong personal opinions or partisan affiliations to influence their forecasts. Instead we wanted them to rely as much as possible on the information that they could collect online, as a group, in a very short amount of time and under high uncertainty. By using their private browsers—arguably dense with cached information about their “digital persona”—we can test whether distance along a large profiling space does indeed result in differential access to information, so that more distant people are more likely to find different information. In other words, we relied on the heuristic “the more different two people are, the less likely they will be to belong to the same online information bubble”. We highly appreciate the reviewer’s comment because it allowed us to bring forward our thinking and implicit assumptions, which before remained under the surface of our main narrative. As correctly pointed out by the reviewer in one of their following comments, what ultimately matters in collective decision tasks is the distribution of forecasts in our group (what we call in the paper, information or “state” diversity, as opposed to “trait” diversity). A new exploratory analysis revealed that indeed larger Euclidean space was inversely related with the correlation of forecasts in our population, particularly after stage 2 (online browsing). In other words, after searching for task-relevant information online (but not before) a relation emerged so that the more distant people were on the multi-trait

¹ <https://www.wired.co.uk/article/tiktok-filter-bubbles>

profiling space the more uncorrelated their forecasts were. This relation stayed until after the social stage (see page 7 below for correlation coefficients).

So, perhaps a better way to think about our diversity manipulation is by considering the scope of our investigation. That is, we are not necessarily interested in drawing final conclusions about an exact axis of diversity (we leave this challenging task to specialists in the field), but we can still inform the discussion on diversity with methods and insights from computational social scientists (and vice-versa inform the discussion on online filter bubbles with rigorous methods and information-theoretical framework using in the diversity literature).

Feature selection. We hope the explanations above clarify the motivations behind our approach. In order to operationalize this intuition (multi-trait segmentation may reflect in information access online and thus affect the ability to solve collective tasks), we apply a common unsupervised clustering algorithm to our large multi-trait profiling space. We used DBSCAN because it is a common data-driven clustering algorithm with clearly defined and interpretable parameters. Regarding the selection of individual features included in our pre-test battery we relied on prior literature on the different aspects of diversity and team performance, which includes common demographics often used to segment populations online. Every dimension selected had a previous connection with some aspect of diversity that was known in the literature. We also included social and relational indicators (e.g. “What is the political orientation of your average friend from 0 (Far left) to 100 (Far right)”) to gauge social and network effects, which have been shown to be relevant for both diversity research and data science (AlShebli et al. 2018, *Nature Comms*).

However, we agree with the reviewer that simply “measuring everything about a person” is not informative unless we can better characterize our experimental variable (diversity as distance in feature space) and the mechanisms by which it affected group performance. We do this in the paragraphs below.

“Average Euclidean distance along those axes to measure diversity can certainly be calculated. The question is whether it means anything.”

AUTHORS: We thank the reviewer once again to give us the chance to clarify our implicit assumptions.

Euclidean distance as a proxy for information. It is useful to notice that it is better to understand our Euclidean trait diversity not as an explanatory variable (E.g., diversity does XYZ), but rather as a tool (a proxy) that allows us to create groups that differ in the underlying information pool that they can collectively tap into online. As noticed by the reviewer below, it is known that in decision making and forecasting tasks, the distribution of errors in the group affects the outcome of the result. One of the rationales behind the use of diversity in team performance is that it is a way to affect the distribution of errors and reduce the amount of bias that a group shows (Hong and Page 2004, *PNAS*, and, more recently, Bendor & Page 2018, *J. Econ & Man. Strat.*). In a new exploratory analysis, we thus measured the correlation existing

between the Euclidean distance between pairs of participants along the profiling space, and the correlation of the forecasts that the same pair produced across the eight forecasting problems. The two measures were negatively correlated (meaning the greater the Euclidean distance the lower the correlation of their answers). Importantly, this relation emerges after stage 2, namely after participants had the chance to forage for information online, arguably using their personal browser (initial: $r=0.12, p=0.38$; Revised: $r=-0.39, p=0.006$; Final: $-0.056, p<0.001$). Thus although “similar” people (ie. a small distance along the Theta space) do not necessarily produce correlated judgments in isolation, their forecasts can become correlated after they search for task-relevant information (even though for a very short amount of time) online. This suggests that using Euclidean distance was an effective proxy to manipulate the error distribution of our population.

Dimensionality reduction. At the same time, we took further steps to better characterize our continuous trait diversity measure (Euclidean distance). As explained, our working heuristic (“more diverse people are less likely to belong to the same filter bubble”) was agnostic about which dimensions of variability were going to be the most important for our task. However, this does not stop us from investigating post-hoc which dimensions were important (namely those which explained large portions of variability in our population) and which were redundant. This approach is very common in other psychological disciplines like in marketing research (Brakus et al, 2009, *J. of Marketing*). We believe this post-hoc characterization is very important, particularly considering that it is difficult to know in advance which segments of the MTurk population are going to engage with a specific online experiment (although recently papers have been published with the explicit intent to describe the MTurk population). We adopt a principal component analysis (PCA) approach and observe that our data shows clear structure. In particular, most questionnaire dimensions did not capture any meaningful variance. Only 5 components explained more than 5% of the variance. In a new analysis, we adopt common methods to find non graphical solutions to the scree plot. According to a parallel analysis, 8 components are selected, and while only 1 is selected if we consider an acceleration approach.

Figure. First ten principal components' explained variance. Although 29 dimensions were included in our original pre-test survey battery, only 5 principal components explain more than 5% of the variance, suggesting strong structure in our data. The limited number of principal components show that most dimensions were either redundant (correlated with others) or showing not enough variance across participants. The red line shows the 5% threshold. The green dashed line represents chance level (1/29th) as suggested by another reviewer.

Non Graphical Solutions to Scree Test

Figure. The acceleration factor in the figure above corresponds to a numerical solution to finding the elbow of the scree plot. It avoids issues in using more subjective estimation methods. It corresponds to the acceleration of the curve, i.e. the second derivative. In our data, the acceleration method suggests that only the first component (that

exploratory analyses showed was highly correlated with the White-non White construct dimension) should be retained. Similarly, the parallel analysis is another empirical non graphical method to find the number of components that should be included to correctly represent the data. According to a parallel analysis the first 8 components should be retained.

These selected principal components corresponded to specific survey items, as judged by the absolute loadings (figure below), instead of being a linear combination of multiple survey dimensions. This made the principal components interpretable. Many original survey dimensions, e.g the ones linked to profession and education did not seem to capture meaningful variation in our population.

We report below the most loaded dimensions for each principal component, for interpretability.

PC1: Race/Ethnicity: White - non White

PC2: Race/Ethnicity: West-East

PC3: Political orientation: Far left - Far right

PC4: Sexual preference: Heterosexual - Homosexual

PC5: Race/Ethnicity: Asian - non Asian

PC6: Personality: Need for cognition score

PC7: Race/Ethnicity: Hispanic - non Hispanic

PC8: Race/Ethnicity: Black - nonBlack

PC9: Age

Figure. Absolute loadings broken down by principal components and original dimensions. High absolute values suggest that a particular original question was highly loaded on a specific component. Notice that for each principal component, one to three questions show the greatest loadings, while all others show values close to 0. This suggests good interpretability of the principal components (see also Figure S14-S16).

Finally, in a new last analysis, we show (figure below) that participant segmentation in the core, inner and outer segments of our sample can already be distinguished on just the first 2 components. Notice that a nice radial symmetry is observed that corresponds to the schematic of Figure 1a in the main paper, with orange dots corresponding to the core segment, the blue dots corresponding to participants in the inner segment, and green participants corresponding to the outer segment. Further notice that inner segment participants are largely overlapping with the core segment (the two composing “homogeneous” groups), while outer segment participants are instead dispersed around the periphery of the distribution (core and outer participants both composing “diverse” groups). Dot size corresponds to whether the participant was used as a treatment for core participants. This procedure follows closely the experimental procedure to randomize diversity pioneered in (Antonio 2004).

Figure. The figure shows the projection of participants' responses to our original full pre-test survey (29 questions) into the first two principal components. A clear radial symmetry can be observed whereby core and inner segments participants overlap more than core and outer segments participants. The figure corresponds broadly to the schematic in Figure 1a.

The conclusion that emerges from this in-depth analysis of our Euclidean measure and its relation with our manipulation of group assignment is the following:

1. Rather than focusing on a single measure of diversity to explain its causal contribution to group performance (a psychological approach), we chose to adopt an instrumental approach (the practitioner’s approach) to see whether high-dimensional unsupervised segmentation can lead to different group outcomes. Our choice was motivated by a well-grounded concern in recent practitioners’ design choices, adopted in AI and related fields.
2. Large Euclidean distance along the profiling space Theta between pairs of individuals is inversely related to the correlation coefficient of the forecasts made by the same two

individuals. Importantly, this relation emerges stronger after the revision stage, namely when individuals have had the time to forage information online. In other words, the smaller the Euclidean distance the more correlated their judgments will be after online browsing. This suggests that our multi-trait diversity manipulation was successful in creating information diversity.

3. Variability along the profiling space is highly structured, meaning that only a few principal components explain most of the variance.
4. Only a few original dimensions (survey questions) show strong absolute loadings for each principal component. This means that rather than one principal component being an unintelligible linear combination of several other original features, we can estimate which original dimensions (which questions in the pre-experimental battery of questions) correlated with a specific principal component.
5. Good radial symmetry is observed when projecting participants' answers in two PC dimensions, suggesting that, as expected by our manipulation, participants belonging to the core segment were indeed closer to participants belonging to the inner segment than to participants belonging to the outer segment. This, combined with the result in point number 2, is also evidence that groups in our "diverse" condition were also more likely to give less correlated judgments.
6. Points 1-5 should reassure the reviewer that, although unconventional, our Euclidean distance measure is theoretically important to investigate, and that it produces visible and interpretable effects on the informational diversity (ie. the correlation in judgments and errors) and thus group accuracy.

“Did you compose your groups also according to other (more traditional) diversity measures to investigate whether your results are robust?”

AUTHORS: We agree with the reviewer that this is an important way to convince the reader that our measure did not bias our results. For this reason we report more prominently in the text a comparison with a more traditional diversity measure. However, our mean Euclidean distance (MED) was not a post-hoc measure that we calculated on each group. Instead, it was used as the criterion to create the group themselves (Figure 1 in the main text). Thus we can certainly compare MED with an alternative measure (as we do in the paragraph below) but composing groups using this alternative measure would require rerun the study entirely.

“What makes average Euclidean distance such an appropriate measure and how it compared to other measures?”

AUTHORS: As mentioned above, to convince the reader that our measure did not bias our results, we report a comparison with another commonly used diversity measure. As it is seen in Figure S12 in Supplementary Information, our MED correlates strongly with the standard deviation ($r=.92$, $p\text{-val}<.001$). Standard deviation is a very common measure to compute diversity that can also be used on multivariate variables. Other measures, such as the Gini

coefficient, are not suitable in this study because they require one-dimensional inputs. We now refer to this analysis more prominently in the manuscript.

Figure. The figure shows the correlation between mean Euclidean distance on the Theta space (multi-trait pre-test survey space) and standard deviation. The two are highly correlated, thus validating our Euclidean distance measure.

List of relevant edits:

- We refocused the introduction to bring forward the motivations behind our operational choices. Before these were only briefly mentioned in the discussion.
- We strengthen the notion that this choice was intentionally agnostic (Ln 41)
- We clarify our experimental question (how does high-dimensional unsupervised user segmentation reflect on the information that participants are able to retrieve online and thus their ability to solve complex tasks together?) (Ln 16)
- We clarify the role of Euclidean measure to produce informationally different groups (Lns 46-54)
- We analyse the relationship between participant distance on the profiling space and the correlation of their forecasts, broken down by different response stages (Ln 122)
- We plotted the mean Euclidean distance against standard deviation as a standard alternative measure of diversity (Figure S12, Ln 60)
- We investigate the structure of the Theta space, and thus what diversity really means in our study using a principal component analysis to understand the variability of our sample along specific components of diversity.
- We show that structure of our manipulation (ie. the clustering of participants in three groups) can already be observed on a 2-dimensional principal component projection of our participant answers to the pre-screening battery. (Figure S17)

The title makes the results sound like dogma but this finding is just one realization under particular circumstances. Other articles have found that diversity can also benefit small groups (Kurvers et al. 2016 PNAS) or that diversity particularly benefits intermediate groups (Kao and Couzin 2014 PSRB) – it all really depends on how you define diversity and what the accompanying circumstances are.

In view of this I would rephrase: “Diversity can promote... and may harm...”

AUTHORS: We totally agree with this point. We changed the title according to the reviewer’s suggestion. We believe the new title better reflects the scope of the paper. It avoids sounding too dogmatic or generalizing conclusions beyond our results. Also, please notice that we now stress the specificity of our claims throughout the paper.

Regarding the papers mentioned, we thank the reviewer for bringing these papers to our attention. Kurvers et al found that different rules should drive the selection of the group answer, as a function of the similarity of their performance. From Kurvers et al: “*The confidence/majority rule outperformed the best individual only when the diagnosticians’ accuracy levels were relatively similar*”. The authors find that when performance similarity is high, taking the answer of the majority or the answer of the most confident individuals improves performance over and above the performance of the best individuals in the group. However, when the performance difference between judges is higher, then taking the best performing individual performs better than majority or confidence based aggregation rules. Please, notice the following:

1. The task used in this experiment was a signal detection task. These are notoriously simpler tasks for collectives, because the distribution of judgments tends to bracket the true value (Armstrong 2001, Krause et al. 2010, Koriati 2012²). However, geo-political forecasting tasks tend to be more complex because of several reasons, including they do not respect signal detection theoretic assumptions, confidence often is inversely related to accuracy (Koriati 2012) making social interaction problematic, judges show correlated information sources that influence the correlation of their errors (see Kao and Couzin 2014, PSRB), and rare events tend to be dramatically influential, which could remain undetected if only looking at raw performance measures.
2. Contrary to Kurvers et al study, in forecasting we cannot rely on the best performing individual, in case of large differences in performance between judges. In forecasting tasks, we cannot adopt this strategy because we do not have access to specific historical records of an individual performance. Pioneers of the field like Phil Tetlock are pushing the field exactly in this direction, by recording long term performance of forecasters. Unfortunately however, for many episodes in modern geo-politics we cannot rely on such data, given that governments and analysts often operate in a highly dynamic social, technological and political context.

² "When are two heads better than one and why? - NCBI."
<https://www.ncbi.nlm.nih.gov/pubmed/22517862>. Accessed 12 Jun. 2020.

For these reasons, we believe that papers that are much closer to our case are the ones by Kao and Couzin (2014 and 2019, *PRSB*), Galesic et al (2018 Decision), Wu et al (2019, *Nature*). We cite these papers throughout the paper to inform the interpretation of our findings.

List of relevant edits:

- Changed the title to better represent the scope of our findings
- Toned down the generality of our claims throughout the paper
- Introduced relevant literature

I often wonder whether diversity in a decision-making context is even a useful term. Error is correlated to a certain degree between individuals and that is the only type of diversity that is meaningful for group performance. Every other kind of diversity (e.g. homophily, demographic factors) is only a proxy for this degree of independence.

You acknowledge on line 75 that diversity is a complex construct but nevertheless then delve into its discussion anyway – without defining diversity. This is not helpful.

AUTHORS: We agree with the reviewer’s judgment and apologize for giving the impression of under-defining our constructs. The reviewer’s comment exactly captures the idea that multi-trait diversity in our work was not to be intended as an explanatory variable (i.e. diversity does XYZ) but as a pragmatic experimental approach to affect information (i.e. via diversity so defined, we want to create groups whose information and error correlation differs). We are very conscious of the fact that demographic and other factors considered were only a proxy for information diversity, but failed to be explicit about it. We agree with the reviewer that trait diversity in a decision-making context is indeed useful only insofar it affects error correlation. In our manipulation, we needed a way to manipulate information diversity (unobservable) without having to assign information to participants ourselves (this would have greatly reduced the generalizability of the conclusions). In this new submission, we thus are more explicit about our reasons and we introduce new analyses showing how multi-trait diversity was indeed associated with error correlation.

List of relevant edits:

- We refocused the introduction to reflect this reasoning. Using Euclidean distance on an arbitrary large profile space is a functional choice that allows us to create different error correlations within the group. It is not an explanatory variable per se.
- We clarify in our introduction that this approach is motivated by our interest in the effect of multi-trait user segmentation on our collective capacity to retrieve and use task-relevant information to guide our collective and private decisions.
- We edited the sentence in question as the new introduction better defines the scope of our metric.
- We better characterize what diversity meant for our study

- We report more plots and analyses of distribution of errors, and their evolution over the successive phases (Figure S2)
- We report a new analysis on the correlation existing between Euclidean distance and forecast correlation (Results *Multi-dimensional profiling* section)

I also wondered about your results in the context of recent findings by Kurvers et al. PNAS. They found that uncorrelated error (i.e. diversity) only mattered if both participants were of similar ability. If one participant is considerably better than the other then there can be no benefit from a group decision. Do you have enough decisions from individuals to measure ability and to see whether this applied to your small-group decisions as well?

AUTHORS: We thank the reviewer for this suggestion as it generated new insights and analyses. We were aware of these findings, originally reported in Bahrami et al (2010) *Science*. Thank to the reviewer's connection, we looked into individual ability (Brier scores) and its distribution within each experimental group to see whether the similarity of group members' performance could help us interpret our results.

As requested by the reviewer, we analysed the diversity/similarity of performance that we observed in our groups across conditions. We calculated initial forecasting performance for each participant as the average Brier error across the 8 forecasting problems (IFPs). Then, for each group we computed the standard deviation of this performance measure across participants. Larger values indicate that a group contains members who are very good (on average across the 8 IFPs) and members who are quite poor. Results are shown in the figure below and reported in the main text. They show that during initial forecasts, people's forecasts are quite similar to each other, typically with a within-group standard deviation (in Brier scores) of around 0.1-0.2. The picture however dramatically changes when people are asked to "forage" for task-relevant information online. Performance diversity (ie. the standard deviation of the mean performance across participants within a group) now increases for small diverse groups but not so much for small homogeneous groups nor for large groups. In other words, when people are allowed to forage information online, members of small diverse groups seemed to end up on a wider performance range. This is likely due to the compound effect of (1) small groups being noisier; 2) trait diversity in our study (as measured by our Euclidean distance) showing a negative relation with the correlation of individual forecasts, meaning that members in diverse groups were more likely to give uncorrelated judgments. On the contrary, large diverse groups did not seem to show an increase of the variability in performance, likely due to lower noise levels (Kao and Couzin, 2014, *PRSB*).

Finally, the same pattern observed in revised forecasts is maintained in final forecasts, that is after social information exchange. Small diverse groups reduce their variability in performance (likely due to interaction and convergence on similar forecasts) but to a much lesser degree than small homogeneous groups (orange). The latter seem to converge to similar performance levels (lower variability) more than other conditions. In other words, and in accordance with our

conflict analysis (now “disagreement analysis”) in Supplementary material, small homogeneous groups show, after interaction, similarity in performance, suggesting that their forecasts were more aligned with each other than compared to similarly sized groups in the diverse condition as well as compared to larger groups.

Figure. Variability in performance (also shown in the main text) for each experimental condition. Notice that small diverse groups tend to decrease their performance similarity after online browsing (revised forecasts) more than other groups. Performance similarity has been shown to positively predict collective intelligence (Bahrami et al. 2010, *Science*, Kurvers et al. 2016 *PNAS*).

As a comparison, we plot in Figure 4 the conflict existing between different conditions and across different stages of forecasting. Notice that, once again, groups across conditions start with very similar levels of conflicting evidence, namely the standard deviation of the forecasts within the group. We rename this conflict “disagreement”, following the later suggestion from the reviewer. Although disagreement seems to increase for all groups in the revised stage (likely due an increase in confidence of one’s initial forecast), small groups in the diverse condition do not recover well, while small homogeneous groups and large groups seem to do much better (less disagreement). Once again, the picture emerging is that our Euclidean distance manipulation, as well as our modularity manipulation, produced observable differences in our experimental groups during the revised stage, which continued the social interaction phase. Such differences seem to have negatively affected small diverse groups while leaving the others largely unaffected.

List of relevant edits:

- Added variability of performance measure in panel 4b

Lines 375-378 The major result described here is supposed to be found in Figure 3b. However, in Figure 3 I could find no label for (b) in the caption.

The same is true for figure 2, I believe.

AUTHORS: We appreciate the reviewer's attention to details and thank them for spotting this mistake. We have corrected the mistake in this new submission.

I also had some issue with the interpretation of the data because I am not sure whether you can conclude from your data that there was greater "conflict" if the standard deviation of diversity was greater. It could also mean that participants with greater diversity had different ways to articulate themselves or that there is greater variance in social dominance to name just a few potential factors. I know that many variables entered your diversity measure but this doesn't mean they were controlled for in the actual discussion.

Cherry-picking the one you like seems a little selective. At best I would speak of "conflict-potential" because you never empirically measured any form of conflict.

AUTHORS: We thank the reviewer for their feedback. We took their recommendation on board and changed the term to a more appropriate one.

We would like to offer a clarification of the concept of "conflict" provided in the previous version. There may have been a miscommunication due to lack of clarity on our side. Our original measure of "conflict" was not the standard deviation of diversity but the standard deviation of forecasts within a group. This measures the spread of the distribution of forecasts, following the intuition that more variability in the probabilities produced by members of a group (e.g. one person says 10%, the other says 90%), entails more conflicting evidence the members of the group have to reconcile when trying to come to a consensus via social interaction. This measure had nothing to do with the Euclidean distance measure, which was our "trait diversity" measure that was used to sort people into groups.

We agree with the reviewer that the term "conflict" may be interpreted in a social-psychological sense (e.g. inter-personal or inter-group conflict). However, here we are only interested in the variance of the forecasts made by a group (purely an information point of view). It is known that such variability in opinions or estimates reduces after social exposure (Lorenz 2011 *PNAS*) and is crucial for many wisdom-of-crowds effects and social failures. We agree that dominance might have differed between groups of different degrees of diversity, but we are not sure about mechanisms translating dominance into a different spread of forecasts.

We have taken the reviewer's suggestion on board and changed the term that may have caused confusion. We preferred to remove any hint towards the term conflict, given the connection of this term with specific interpretations. We opted for the more general term "disagreement" which better captures our idea of conflicting evidence/beliefs that the group has to reconcile via social interaction. Please notice however that a more neutral term may also be considered. "Disagreement" could also be named as "diversity of opinions", depending on whether we want

to give it a positive or a negative connotation. For example, Lorenz and colleagues (2011, PNAS) talk about opinion diversity, but they essentially refer to a very similar construct, namely how spread apart are the individual forecasts within a group of forecasters. In Lorenz’s paper (an estimation task!), “diversity of opinion” is more appropriate than in our paper. Indeed, while in their paper diversity of opinions is associated with the greater possibility of bracketing the true value (Hong & Page (2004) PNAS), in our task the same measure means that people have not fully reached a consensus with their peers, which in turn may lead to lower performance in our task. We thus use the term “disagreement” to reflect this subtle difference. We replot this measure below and in Figure 4 in the main text.

Figure. Disagreement is quantified as the standard deviation of forecasts within a group. Notice that small diverse groups tend to have larger disagreement than other groups, especially after the revised forecast stage.

List of relevant edits:

- We changed the term “conflict” to “disagreement”, as this better captures our meaning.
- We added the full disagreement distributions to Figure 4

Reviewer #2 (Remarks to the Author):

The manuscript has been substantially improved. I would like to thank the authors for their dedication engaging with all my previous concerns. I believe that, in the current version, the results and framing of the paper are much stronger compared to the first submission. I only have two final -relatively minor- comments, which should be taken as suggestions to make the manuscript even more clear (comment #1) and to justify the selection of five components in their PCA (comment #2).

AUTHORS: We were delighted to see the reviewer's feedback and appreciate their time and insight. We have replied to their further comments below.

1) Regarding the framing of the paper: The authors mention in several parts that they adopt the "well-established framework of social learning" (line 28, abstract) and that they look at diversity through the lens of "social learning and network science" (line 117). Even in the abstract, they suggest that they import knowledge from cultural evolution and ecology, but it remains unclear exactly what specific knowledge from those fields are informative to the current study. I would like to suggest the authors better explain exactly what they mean by "the framework of social learning" given that there are many theories/models of how people learn in social settings.

AUTHORS: This is certainly true, the framing of social learning was much more prominent in previous versions of the manuscript and it is perhaps poorly explained in the current version. We refocused our abstract and introduction accordingly. We mention this literature in the paper because it offers a quantitative approach to information and error correlation in groups and networks, and thus can help us frame our findings. However, we de-emphasise the centrality of social learning theory for the main narrative of our work. Instead, we focus more on the group composition aspect, which other reviewers suggested needed more unpacking.

We want to offer a clear explanation to the reviewer of what we meant by adopting the "well-established framework of social learning". Social learning theory and network science have made great progress in offering very clear mechanistic descriptions of information and error processes affecting groups and networks. The two slightly differ, the former is typically focused on dyadic interactions and copying behavior in groups, while the latter is focused on the effects of network structures on different problem-solving abilities, like navigation of complex problem/solution spaces, social contagion and opinion cascades. However, both research programs find convergent evidence that, under conditions of uncertainty or costly learning, copying others allows individuals to better solve complex problems. Both in groups and networks, if solutions are allowed to spread from one individual to the other via several learning mechanisms highlighted in that literature (eg. by copying prestigious or high performing individuals, copying the majority and herding, or direct vertical and horizontal transmission), then the costs to individuals diminish, while group performance tends to remain unaffected. Importantly however, the correlations in judgments and errors that interaction or social learning entail require a mechanistic understanding of information processes and more quantitative models. In the previous version of the paper, we suggested that this view may be beneficial for the scholar interested in group diversity and performance because it allows them to make new predictions and observe new effects. We highlighted that one such unexpected effect is the interaction between two factors (group size and diversity) that before were seen as independent.

List of relevant edits:

- Refocused abstract and introduction
- Refer to the social learning literature findings of the complex interplay among several factors affecting group outcomes (less prominent in diversity research), e.g learning strategies, task complexity, group size, modularity and network communication structure.

2) I appreciate and liked very much the PC analysis performed by the authors where they selected five components explaining a large proportion of the variance (~90%). Given that this is a 29-dimensional space and the remaining 24 dimensions explain ~10% of the variance, however, one could still have more than five meaningful components. For example, if the sixth component explained 9% of the variance, then this would be approximately 3 times larger than what one would expect simply by chance (roughly $1/29=3\%$). To formally select a given number of components, the authors could perform a random permutation analysis and show that from the sixth (or seventh, or eight,...) component the remainder amount of variance is consistent with what is expected by chance. This would help justifying the selected number of components in their analysis.

AUTHORS: We thank the reviewer for this very insightful comment. We must admit we had not thought about this nuance. We took this suggestion on board and reported our findings below. We ran standard non-graphical solutions to the Scree test, like parallel analysis and acceleration factors. To summarize the results below, the former returned a single component, while the latter returned eight.

Figure. The acceleration factor in the figure above corresponds to a numerical solution to finding the elbow of the scree plot. It avoids issues in using more subjective estimation methods. It corresponds to the acceleration of the curve, i.e. the second derivative. In our data, the acceleration method suggests that only the first component (that exploratory analyses showed was highly correlated with the White-non White construct dimension) should be retained. Similarly, the parallel analysis is another empirical non graphical method to find the number of components that should be included to correctly represent the data. According to a parallel analysis the first 8 components should be retained.

We report below the original questions that showed the highest absolute loading on each component:

- PC1: Race/Ethnicity: White - non White
- PC2: Race/Ethnicity: West-East
- PC3: Political orientation: Far left - Far right
- PC4: Sexual preference: Heterosexual - Homosexual
- PC5: Race/Ethnicity: Asian - non Asian
- PC6: Personality: Need for cognition score
- PC7: Race/Ethnicity: Hispanic - non Hispanic
- PC8: Race/Ethnicity: Black - nonBlack
- PC9: Age

Similarly, we expanded the Supplementary information material with a new figure showing the absolute loadings for each question on each principal component. It can be observed that only a few questions corresponded to each principal component, suggesting that principal components were interpretable.

Figure. Absolute loadings broken down by principal components and original dimensions. High absolute values suggest that a particular original question was highly loaded on a specific component. Notice that for each principal component, one to three questions show the greatest loadings, while all others show values close to 0. This suggests good interpretability of the principal components (see also Figure S14-S16).

List of relevant edits:

- We expanded the PCA section in supplementary material
- We included more components in our SI figures.
- We report in the main text these results more explicitly and in depth

Reviewer #3 (Remarks to the Author):

Following revisions, there are still substantive issues with this paper that need to be addressed. The claims made are suggested to be generic, but there are many aspects of this work that are problematic, being based on very specific aspects (as detailed below) which makes it not possible to generalise the results in the way the authors claim. They need to substantially tone down their claims and directly address the specificities of this work, as outlined below.

AUTHORS: We are grateful for this second opportunity to appropriately address the reviewer's concerns. In this new revision, the reviewer will notice that we have substantially toned down the generality of the language across the whole paper, starting from the title itself. As a result the paper has greatly improved. The narrative and language better reflect the specific nature of the task and the generalizability of our findings. We also address the individual concerns of the reviewer as reported below.

List of relevant edits:

- We changed title to better characterize the limited scope of our study
- We refocused the introduction, reducing the role of multi-trait diversity as an explanatory variable (the generality claim observed by the reviewer) and highlighting its functional role in creating differences between individuals in information/error correlation.
- We toned down or removed all preregistration claims. We provide information to the interested reader on how to access our pre-registration material to learn about our hypotheses before data collection.
- We highlighted the fact that many analyses are exploratory in nature.
- We discuss the value of exploring the data to (a) gain convergent evidence (from multiple measures) on likely mechanisms affecting the patterns observed in our data; (b) generate new hypotheses for future investigation.

The paper begins with a bold claim about the role of diversity and group size in promoting collective intelligence. Diversity is, inherently a broad term, and it's unclear what it means in the

context of DBSCAN-sorted groups that are forced into three clusters. Are these natural clusters? If not, how might other latent and un-captured structure interact with group size? The use of PCA to uncover the major axes of variance is useful, but it remains unclear what diversity means in the context of this study, and less so how these results could be considered general.

AUTHORS: We thank the reviewer for giving us the opportunity to address this point. We approach this issue on a number of grounds. First, we refocus our introduction and discussion to better explain the motivation behind our unsupervised clustering approach to diversity, and its relevance for computational social science. Second, we stress the fact that we use Euclidean distance only as a tool (a proxy) to sort people into groups that are likely to differ in informational diversity, we don't claim that this measure is intrinsically meaningful, generalizable or the 'correct' measure of diversity. Third, we characterise the informational meaning of Euclidean distance in our study by looking at the relation between this metric and forecast correlation between pairs of participants. Fourth, we further characterise *post-hoc* the multi-trait survey data this metric is based upon, so to understand variance in our experimental population. Principal component analysis was expanded to include more objective non-graphical selection criteria for principal components and the relation between principal components and group assignment in our experimental manipulation. Fifth, we tone down the generality claims of the paper (starting from the title and abstract) to highlight that these results apply only to a specific task and context, and caution should be adopted in generalizing our results to other aspects of diversity. Sixth, we investigate the correlation between Euclidean distance and alternative measures of diversity, and provide further analysis of disagreement (previously "conflict") patterns in groups, raw forecasts and error correlation and differences in performance emerging when people are allowed to browse information online (revision stage). We provide a more detailed account of these points below.

“Are these natural clusters?”

Multi-trait clustering. The three clusters found using DBSCAN are not natural clusters that signify distinct categorical differences between participants, but serve the only purpose to 'manipulate' (i.e. randomize) trait diversity. This is clarified in the revised introduction. We assume that people naturally lie on a continuous distribution of characteristics in an arbitrary large feature space. The motivations for choosing a large feature space instead of a single dimension of diversity are made center stage of the introduction in this new submission. This design choice lies in our interest to understand collective information foraging and decision-making in natural online environments. Recommendation systems are known to affect both content and search results online. These systems' design principles rely on clustering users' populations based on large profiling spaces, often including demographics, relational and behavioral indicators. Such segmentation has been shown to affect people's behavior in a number of ways, from content retrieved³ to behavioral shifts (Epstein et al 2015 *PNAS*). Although individual shifts in behavior are certainly important to investigate, an equal important

³ <https://www.wired.co.uk/article/tiktok-filter-bubbles>

question that we want to bring to the attention of the reader (and try to tackle in the paper) is that such differential information access based on multi-trait user segmentation may affect our collective ability to retrieve decision-relevant information online and use it to converge to accurate collective decisions. In our task, the use of online browsing to retrieve task-relevant information, coupled with the low prior knowledge of the forecasting problems, was likely to affect the information that a group can collectively tap into when exposed to online content. We notice that in a digital space where behavioral, psychological and demographic traits are transformed into differential access to information (e.g., the bspoken “filter bubble”), the tension between demographic and informational diversity, often invoked by psychologists, may not apply or may need to be revisited. Our approach thus tries to reconcile the psychologist approach (usually manipulating single axes of diversity) with common practices in AI and data science aimed at segmenting users populations based on large numbers of known features (including demographics and relational indicators) to modify behavior.

To answer the reviewer’s question more directly, we did not assume before data collection that the three clusters reflected any intrinsic underlying cognitive or demographic constructs. Instead they solved a purely functional purpose. In order to manipulate within-group trait diversity, we need to randomly assign 50% of our population that is equal in expectation (our test subjects, namely core participants), to interact with either 25% most similar or most dissimilar others (our treatment subjects, namely inner and outer periphery participants). DBSCAN offered a simple data-driven approach to this problem. It offered the advantage to rely on a small number of parameters that were easily interpretable. Given the difficulty in randomly assigning diversity (by definition we cannot randomize demographics nor cognitive measures), we need to use a subpopulation as an experimental treatment for our core subpopulation. We are not the first ones to use this approach. For example Antonio (2004), one of the pioneers of this method, used black and white students as a treatment to a second group of white students who were randomized to interact with them. This method allows the experimenter to use the natural distribution of characteristics in the sampled population to randomize a subset of subjects who are in expectation identical into groups that differ in their trait composition. This is a very useful tactic to infer causal relationships between diversity and other cognitive or behavioral indicators (in our case, forecasting accuracy and disagreement).

“If not, how might other latent and un-captured structure interact with group size?”

Once again we thank the reviewer for asking the right questions. Regarding the existence of other latent un-captured structures, this is exactly why the use of randomization is so crucial (and thus the use of DBSCAN is so needed) and why we opted for an experimental methodology. Randomizing a subset of participants and focusing our analysis on them allows the experimenter to alleviate concerns (assuming a large enough sample) about other covariates because randomized participants will be, in expectation, identical. Any unobserved latent variable is thus expected to be orthogonal to our manipulation.

Euclidean distance as an information proxy. Notwithstanding these technical details, we notice in the main text that

“For the scope of our paper however, the specific definition of multi-trait diversity is not as important as its functional value of using trait diversity to influence information diversity and error distributions in online groups.”

Although we do not believe that people in the core segment were qualitatively different from people in the inner segment (as also addressed in the PCA section), we think our experimental manipulation created measurable information differences when such groups were allowed to forage information online and debate their views with each other. Thus, for our purposes, the role of the demographic characteristics is purely instrumental to create groups that are likely to differ in terms of the underlying correlation of their forecasts/errors. As pointed out by Reviewer 1, what really matters in this decision-making task is “only” the information pool and correlation of judgments among individuals (what we call “information diversity”). All dimensions in our pre-test survey (like demographics, personality and cognitive measures) were used to create an effective experimental information treatment. We worked under the assumption that the more “different” two people are on an arbitrary large profiling space, the less likely they will be to belong to the same information bubble. Of course, we did not pick these dimensions at random, but based on previous literature as well as common features typically collected about users online (so called digital traces) to provide customized search results and/or content recommendations. These typically involve common demographics but also include social-relational indicators capturing variance in the network the focal participant is embedded in. For this reason, as well as for the fact that relational indicators were also found to be important in the group diversity literature (AIShebli et al. 2018), we included in our pre-test survey not just questions with an individual locus (“What is your political orientation from 0 (Far left) to 100 (Far right)”) but also with a relational locus (“What is the political orientation of your average friend from 0 (Far left) to 100 (Far right)”).

“The use of PCA to uncover the major axes of variance is useful, but it remains unclear what diversity means in the context of this study”

The informational meaning of Euclidean distance. Prompted by the reviewers’ suggestions we investigated the deeper informational meaning of our agnostic multi-trait Euclidean distance measure. We were very glad to have our work assessed by three insightful reviewers because it pushed us to carry important exploratory analyses that reveal important effects in our data. These analyses are now an important part of the new submission and offer an insightful perspective on the effects of our multi-trait manipulation on a group’s information. To clarify the possible mechanisms underlying our results, and to directly answer the reviewer’s concern about “what diversity means in the context of this study”, we analysed the relationship existing between the Euclidean distance metric (used to assign individuals to groups), and the correlation existing between forecasts of pairs of participants. Our working assumption was that the more dissimilar two people are on a large number of features, the less likely they are to

belong to the same online information silo. If this was indeed the case, we should expect an inverse relationship between the two. Indeed, we find that, between pairs of participants in our population, a negative relation exists between distance on multi-trait space and forecast correlation. Importantly, this negative relation emerges after the Revision stage (but is not present during the Initial stage). In other words, after people had the chance to ‘forage’ for task-specific information online, their multi-trait similarity/dissimilarity predicts their forecast correlation. Thus, the meaning of diversity that emerges is one where trait distance in arbitrary large (and relatively agnostic) profiling spaces produces a significant difference in the information diversity that a group can collectively retrieve or “forage” online, which in turn affects the errors that a group makes in complex forecasting tasks when such information is used to make a prediction. We report here and in the first paragraph of the Results section the correlation coefficients found for each stage of forecasting (Initial: $r=0.12$, $p=0.38$; Revised: $r=-0.39$, $p=0.006$; Final: $r=-0.056$, $p<0.001$). We also provide further controls by correlating our Euclidean distance measure with more traditional diversity measures with multi-dimensional inputs (eg., standard deviance).

Post-hoc principal component analysis. Although we were agnostic on which aspects of diversity were important for our task, we can still characterize *post-hoc* the variability of our population on the large feature space considered. This allows us to zoom in on specific dimensions that were likely to be picked up by our DBSCAN algorithm. Kindly prompted by another reviewer, we further expand our principal component analysis. As before we find that most dimensions were highly redundant or showed little to no variance. We include an acceleration and parallel analysis which represent non-graphical solutions to the scree plot produced. This offers a more objective way for us to pick the number of principal components. According to the acceleration metric (corresponding to a numerical solution to the “elbow” in the scree plot), only one principal component emerges. According to a parallel analysis instead, we select the first eight principal components. We have expanded Supplementary material accordingly. Importantly, selected principal components tightly corresponded to precise ethno-cultural and socio-political dimensions, like race and political orientation (please note the absolute loadings plot in our response to Reviewer 2). This nicely fits with the idea that people who were distant along the principal components of the multi-trait profiling space were also more likely to collect different information when allowed to browse task specific news. We know that search recommendations and content customization are often tailored around these specific features. Finally, to close the circle, we link the principal component analysis to our DBSCAN manipulation. We add a supplementary plot showing a low-dimensional projection of all our participants, divided by their assigned segment (core, inner and outer). A nice radial symmetry is observed, suggesting that, already when considering just the first two principal components, core individuals were closer to inner segment individuals than outer segment individuals. This allows us to understand the likely axes of variation in our experimental groups, thus better characterizing our unsupervised clustering procedure, DBSCAN, and the mechanisms through which our experimental manipulation affected the availability of task-relevant information in the group.

“how these results could be considered general.”

Generality. We agree with the reviewer that the previous draft was too bold in its claims and gave a false impression that our results can be used to draw generalizable conclusions to diversity and collective intelligence across the board. We apologize for giving this false impression as we believe it missed the point that we were trying to convey. In this new resubmission, we are more careful about generalizing our results to new contexts other than complex forecasting tasks under conditions of uncertainty and time pressure. Accordingly, we changed the title of the manuscript to reflect the limited scope of our claims. Rather than collective intelligence *tout court*, we focus on collective accuracy in real-world forecasting tasks under the conditions specified above (uncertainty and time pressure). Although more limited in scope, great value lies in studying such tasks because they can approximate certain aspects of real-world settings.

Finally, we are more explicit about the fact that we do not want to provide conclusive evidence on diversity and performance (i.e. we do not want to “solve” the diversity debate). Instead, we want to draw an exploratory connection between the type of information segregation that is often debated in the contemporary computational social science research program (interested in emerging effects of information processes at the collective level) and group diversity as studied in the lab by psychologists (often focused on single dimensions of diversity and experimental methods). We believe the value of our approach lies in cross-pollinating these two methodologies and hope we have provided enough evidence throughout the paper to support it.

List of relevant edits:

- We refocus the introduction and discussion so make the motivations behind our design choices clearer. These were present in the previous submission, but had previously a minor role.
- We remove bold and general claims about diversity and collective intelligence. We clearly spell out our notion of multi-trait diversity and information diversity, as well as our definition of collective intelligence.
- Similarly, we clarify the findings may have generalizability only within geo-political forecasting tasks, under conditions of uncertainty and time constraints.
- We edited the title accordingly.
- We clarify that segmentation in core/inner/outer does not reflect natural clusters but is an arbitrary segmentation so as to be able to manipulate (ie., randomize) trait diversity.
- We also clarify that we use multi-trait diversity only as a proxy to manipulate information access. We provide evidence that sorting groups based on our unsupervised clustering algorithm indeed affect forecast correlation among people.
- We expanded our principal component analysis thanks to suggestions from Reviewer 2.
- We visualize a low dimensional projection of our participant segmentation.
- We provide distributional plots of performance diversity across conditions, as suggested by Reviewer 1

In addition there are issues with the notion of collective intelligence in this work. It is understood that the questions were selected as part of a broader project, yet the questions asked surround fairly obscure geopolitical events that we find it unlikely could be driven by domain knowledge of MTurk workers. For example, workers would need to know what a Loya Jirga is, why they convene, and the likelihood of events that would prompt such a convention in Afghanistan. It is very difficult to believe the initial answers for these questions were based on prior knowledge for all but a very few select participants. With the *very* brief period to search for information, it's also difficult to imagine they would go far beyond simply knowing what the question was asking. Collective intelligence requires some (albeit noisy) domain knowledge. Even if MTurk participants (unbeknownst to me) have a very deep knowledge of obscure geopolitical events, it is important to note that is a very restricted definition of collective intelligence and the title would be better worded as "forecasting geopolitical events" than "collective intelligence".

AUTHORS: We appreciate the reviewer's feedback and think the point they raise is an important concern that needed to be addressed in the main text. Their comment prompted further thinking about the underlying mechanism of our findings and generated a new analysis that we are happy to share. We thank them for this opportunity. We describe our thoughts in detail below.

Specificity of the task. We agree that the task was highly domain-specific and apologize if the previous wording using the general term of "collective intelligence" gave the wrong impression to the reader. We have now reworded all parts of the text that sounded too general, starting from the title. We limit the scope of our findings to complex geo-political forecasting, under conditions of high uncertainty and time sensitivity.

Stimuli. We also agree that the concern about the type of forecasting problems selected may be shared with many critical readers. And we, as authors, debated it too when designing the task. Due to its importance, in this new submission, we explicitly tackle it in the main text, explaining motivations for choosing such stimuli as well as their scientific value. We provide the following arguments.

First, we are pleased that the reviewer appreciates that the pool of tasks these eight ones were picked from were independently chosen by a national program due to their informativeness to assess forecasting ability. IARPA selected these forecasting questions likely because they were extremely difficult to answer with traditional methods and reflected realistic forecasting problems that intelligence analysts face on a daily basis. Second, from an experimentalist point of view, we firmly believe there is scientific value in unknown ("obscure") geo-political events exactly *because* these topics were obscure. This is because forecasting judgments were less likely to be biased, in one direction or the other, by personal opinions, private information or as a result of political partisanship. On the contrary, more relatable questions (e.g. domestic politics, better

known geo-political events or simply general knowledge questions) would have introduced a host of unwanted confounds that we were determined to keep at bay.

For example, on the one hand, general knowledge questions have been already investigated in the literature and are known to benefit from social interaction because initial guesses tend to bracket the true answer (Navajas et al 2017) or can be inferred by the group pattern of responses (Prelec et al 2017⁴). They are however scarcely generalizable to more complex decision-making in more ecological settings, where ground truths are dynamic or not yet determined.

On the other hand, familiar geo-political matters (e.g. domestic politics) come with a host of progressed knowledge, opinions and political biases that is beyond the scope of this paper. Finding differences in forecasting performance between groups on such a small number of issues would have been difficult to attribute to our manipulation rather than simple political partisanship or being up to date with daily news.

The selection of obscure stimuli then, particularly under great time pressure and uncertainty, had the effect to isolate the ability of the group to collectively search a vast information space that each individual could not have searched alone. The ability to quickly filter out irrelevant information and gather evidence in favor of a decision seems to be at the heart of intelligence agencies' goals as well as an important indicator of intelligence in general. Furthermore, success in this type of task is easier to attribute to experimental manipulation than to prior knowledge.

Finally, we agree with the reviewer that “*collective intelligence requires some (albeit noisy) domain knowledge*” and we were genuinely interested in whether this was the case in our study. We reasoned that if people's judgements were based on pure noise (i.e. no domain knowledge whatsoever), then we would expect accuracy close to 50%. This is true to some extent in initial forecasts as one would expect from the difficulty of the questions asked. As detailed in our signal detection theoretical analysis below, initial binary judgments were unbiased to one or the other event (happens vs. does not happen); Figure S2 also shows that both initial and revised individual Brier scores bracket 50% (random guessing). Yet the forecasting scores found in stage 3 were quite low (meaning accurate forecasting), suggesting that some noisy information was collected online (or was known in advance, albeit this scenario is arguably less likely as the reviewer points out). Social interaction seemed to have the effect of improving the signal to noise ratio between what was known by individuals before interaction and what was known/believed afterwards.

Labelling of chat data. To put quantitative measures to these observations, we further investigated conversations of each group to understand how many groups indeed had some

⁴Dražen Prelec, H. Sebastian Seung & John McCoy (2017). A solution to the single-question crowd wisdom problem. Nature.

domain knowledge of the issue and how many did they completely guessed the answer (or barely understood the question when given the very brief search period. We manually labelled the contributions of each participant for each question and in each group. As hand labelling may be biased by our perspective, we provide the code that can run on the chat logs so that anyone can manually label our data. We found that for all groups the majority of participants had some informed knowledge of the context of the question for the majority of questions. A very conservative estimate suggests that the decimal proportion of participants with some knowledge (averaged across groups) is $M=65\%$, $St.D= 0.04$. Naturally, it is impossible to tell whether this knowledge was known prior to the experiment or collected during the revision phase. We thank the reviewer for prompting this informative analysis.

Generalizability. We agree with the reviewer about the issue of generalizing our results to collective intelligence *tout court*. This misunderstanding was caused by our poor wording. We did not intend to convey this meaning. Although virtually every paper on collective intelligence will have to use a particular task, we think there is value in toning down general claims and understanding the specific effects of our manipulation on the specific task and measures considered. The reviewer will notice that in the new submission, many strong claims were toned down. We draw the reader's attention several times throughout the paper to the need to interpret our results with caution. We think it is now clearer from our writing that our goal is using a realistic complex task to approximate similar tasks in the real online world. We think the value lies in the fact that many papers that often cite collective intelligence use somewhat contrived tasks (like guessing a correlation or estimating the number of beans in a jar), which are not representative of real-stakes complex decision problems in the wild. However, framing any decision problem in terms of information processes allows us to draw conclusions that are hopefully more general. In this new resubmission we tried to modify the text accordingly, drawing attention to the specifics of the task as well as providing more details on the likely information processes underlying our groups. For example, we added a new panel to Figure 4 to show how within-group performance diversity (an important factor in collective intelligence, e.g. cf. Bahrami et al. (2010) *Science*) changed across conditions. This informational perspective allows to better interpret the phenomena under consideration in light of past literature both in the collective intelligence as well as the group diversity research programs (e.g. Mannix et al. 2005).

List of relevant edits:

- We changed the title to better characterize the scope of the paper.
- We changed the introduction to clarify this study's definition of collective intelligence (collective forecasting performance of individuals with little information to collectively search a large solution space). We stress that the forecasting problems used are (a) real world; (b) complex; and need to be solved (c) under extreme time pressure and (d) uncertainty. We clarify the features of the task as described to the reviewer.
- We directly tackle the reviewer's concerns in a paragraph of the Discussion

- We tone down claims of generalizability, and warn caution to the reader in interpreting our findings. Similarly we emphasise the importance of replicating them in future studies and under similar conditions.
- We add to OSF data and code to manually count the number of times groups had relevant knowledge of an IFP at the time of the group discussion.
- We add more details and raw plots to better characterize the information processes underlying each condition.

We are left with a claim the DBSCAN-defined diversity interacts with group size to impact the ability of MTurk participants to forecast unlikely (most did not occur) and obscure geopolitical events. The titles' overly bold claim is, therefore, most likely unwarranted.

AUTHORS: We thank the reviewer for drawing our attention to the importance of toning down our claims. Accordingly, we edited the title, toned down strong claims in the introduction and discussion, and provided better scope for the interpretation of our findings.

List of relevant edits:

- We changed the title and tone down bold claims in the introduction and discussion
- We clarified throughout the paper that we are only interested in understanding complex forecasting in realistic geo-political forecasting tasks, under conditions of high uncertainty and time sensitivity.
- We are explicit about the exploratory and preliminary nature of our results
- We explicitly say that pre-registration did not include analysis

Further, given the ratio of events in the questions that did not eventually occur, it would seem to be inherently impossible to understand whether this represents accuracy versus a simple previous bias to choosing "no". Any claims of collective intelligence could equally be replaced with "preferring to answer 'no'". It is understood that this arose from an inherent constraint of asking prediction questions about real events, but note that this challenge of experimental design does not ameliorate the limitations to the validity of the conclusions. This possibility needs to be addressed directly, as does its consequences.

AUTHORS: We appreciate the reviewer's concern, and are happy to address it in details. We provide a motivation for our design choice and a signal detection theoretic analysis to alleviate this very valid concern. We provide these reasons both to the reviewer and in the main text.

Design choice. As the reviewer points out, the outcome of the forecasting problems could not be anticipated at the time of the experiment. This is the crux of using real world forecasting problems. The fact that the events would have not occurred was not known at the time of the experiment. Choosing real forecasting problems was done to completely avoid introducing biases on the side of the experimenter or of the participant. As neither the experimenter or the participants could know the answer in advance, the task was not about looking for the ground truth (the ground truth was by definition not determined at the time), as it is often the case in

estimation tasks, psychophysics, and general knowledge tasks. Instead the task was to gather evidence (any information) that might be relevant to make the best informed guess. Rather than a defect in our study, we believe this feature captures the gist of forecasting, routinely performed by analysts and intelligence staff across the globe (as well as non-experts). Complex forecasting seems to be a much harder task than say estimation tasks, traditionally used in collective intelligence studies. The reason is that in estimation tasks (as well, to some extent, general domain knowledge tasks) the distribution of guesses does not, or it is not assured to, *bracket* the truth (Armstrong 2001). Instead correlated errors can emerge due to paying attention to the wrong information, or using the same information sources that other group members are using.

The fact that most events did not occur is not uncommon in forecasting, although not necessarily expected. Rare events are often the most consequential and impactful, exactly because they are deemed unlikely. Rare events are by definition difficult to account for in stochastic models and human judgments, often because of a lack of useful information or data. Although difficult to predict, and perhaps because of this reason, they are often highly impactful on matters of national and global security, economics, politics as well for individual lives. Being able to accurately forecast rare events resides at the heart of good forecasting. In Bayesian terms, by predicting a rare event we gain the most information. Einstein's predictions during the 1919 eclipse helped establish relativity theory exactly because they were highly improbable. Predicting a rare event shows the ability to use very few data points to extrapolate trends. Forecasters in our study had to use very limited information to come up with precise forecasting probability measures, under extreme time pressure.

Response bias analysis. Notwithstanding our design motivations, the reviewer raises an important concern, regarding the use of a heuristic, instead of actually making a prediction about the event. This may be problematic if participants had an initial bias toward giving as their independent answer a forecast close to 0 (or at least below 50%). A generalized adoption of this heuristic would have produced a distribution of forecasts within-groups that was largely below 50%. Social interaction in this case would have simply made the distribution more extreme and thus more accurate on average in this specific example. Luckily, we can address the concern regarding whether the human strategy was reflective of a "prefer to answer no" heuristic. To test this hypothesis, we conducted two analyses showing that this was not the case in our data. We find that participants seemed to deem events as equally likely to happen or not happen as described below.

We plot below (and in Supplementary information) the distribution of individual responses for each of the eight questions across participants. On the x-axis we observe the initial forecast, in the range from 0 to 100% probability of the event in question occurring. Although in seven out of eight individual forecasting problems the event did not occur, this was not reflected, at first glance, in the distribution of forecasts provided by participants. Instead, forecasts seem to be randomly spread across the whole domain from 0 to 100 (the entire probability scale). Please notice that we provided monetary incentives for people to minimize their Brier scores. This, in

addition to the fact that the Brier score is a ‘proper’ error measure (that is, it rewards truthful responding), indicates that the distributions below likely reflect the truthful original judgments of our participation pool. For example, it is unlikely that people had a general tendency to believe’ the event was unlikely, but report a forecast randomly distributed around 50%.

However, we concede that we need a more refined statistical tool than simply plotting the distributions. For this reason, we used a signal detection theory approach to precisely quantify the degree of bias of individual answers toward one or the other direction. As the reviewer is certainly aware, signal detection theory represents the standard method to analyse different aspects of decision, especially in binary judgements like the current one. We computed the criterion c , measuring the bias towards responding yes or no with a signal detection theory analysis. We tested (Wilcoxon test) the distribution of values for each participant during the initial response phase and found it did not significantly differ from 0 (meaning no bias).

Figure. Signal detection theoretical analysis of participant's bias (criterion c). Although seven out of eight forecasting problems had negative outcomes (did not happen), participants' forecasts did not show shift in bias. In other words they were equally distributed across the probability spectrum, indicating that they were not knowledgeable of this bias.

The plots and the analysis above suggest that we have no evidence to believe that participants adopted a “preferring to answer no” strategy. This is especially important, because it shows that whatever forecasting accuracy improvement we observe in the results is not due to an initial bias toward adopting a heuristic strategy. Similarly, no evidence was found of a bias in one direction or the other when looking at revised forecasts, namely after participants collected more information online. Instead, responses gravitated to the lower probabilities as a result of the interaction between people within a group. Only after people have the chance (although under time pressure) to collect information from their unique point in the digital sphere and debate it with others, do they indeed improve their prediction accuracy by shifting their forecasts, both private and collective, toward the correct end of the spectrum (which was 0 on 7 / 8 questions). Finally, it is not clear (although possible in principle) how using a “preferring to answer no” heuristic may have produced the patterns of results observed in figures 2 and 3 in the main text.

List of relevant edits:

- We introduced forecast distributions and signal detection theoretic investigation of the individual forecasting data in supplementary material
- We reference the relevant supplementary material in Table 1 of the main text.
- We discuss the concerns raised by the reviewer in the Discussion.

Beyond this, it is concerning the extent to which this paper relied on loose pre-registration and abundant researcher degrees-of-freedom. It would appear to any skeptical reader that this

paper may be a result of HARKing a noisy dataset. We lack simple things like plots of raw brier score (not partial residuals) for each condition, and other useful information to make informed decisions. A plausible alternative explanation is that the results are an artefact of bias and convergence across conditions and have very little to do with collective intelligence or diversity.

AUTHORS: We are sorry if our manuscript gives the impression of having followed bad scientific practices. We have put a lot of effort into preventing p-hacking and post-hoc interpretation of the results and we would like that to show as much as possible in the manuscript. We tackled this issue on two grounds. First we remove all strong claims of pre-registration (starting from the abstract) and specify in the introduction “*Analyses were not pre-registered.*”. Second, we more carefully explore the data and provide raw Brier score plots as requested by the reviewer to provide converging evidence on possible underlying mechanisms. A detailed description is provided below.

Pre-registration. In the current version, we have eliminated all claims regarding pre-registration of analysis, including from the abstract. For full disclosure, we provide a link to the pre-registration material to the reader who may find some value in it. We also provide information about what patterns of behavior we expected from the data before collecting it (clearly spelled out in the Table in one of the comments below). With this information the reader can make an informed decision regarding the weight they want to give to each result. For example the hypotheses spelled out before data collection suggest that Table 2a and 2d, and Table S14 test our main hypotheses (roughly corresponding to the three paragraphs of our Results section), while other results are entirely exploratory. We clarify these nuances in the main text.

Exploring the data. In parallel, we highlight in the main text that most of our analyses are only exploratory. We appreciated the comments of the reviewer that prompted us to reframe and tone down the scope of the paper. We emphasise with the reviewer and the reader the importance of exploratory analysis for our goals, namely offering an empirical approach and preliminary evidence to understand the effects of user segregation online for collective problem solving. Exploratory analyses in this context allow us to better understand macroscopic patterns in the data and thus formulate hypotheses for future studies.

We would like to offer convincing evidence to the reader that our results offer insights in group decision processes in online digital spaces. We are not sure we understand the meaning of convergence across conditions and how it may have consistently produced the coherent pattern of results observed in the experiment. Thus we try to answer to the best of our ability. We believe our results are the result of stochastic effects on group information that our multi-trait manipulation created, and not just the result of bias. This is in part due to exploring the data using different means and finding consistently a disadvantage of small diverse groups, but not of large diverse groups or small homogeneous groups. Notice that we tested multiple small diverse groups instead of a single one, so it is also difficult to explain the results in terms of random variation. In this new resubmission we do a better job at reporting a range of exploratory

data analysis that tell a coherent story. By plotting multiple indicators (performance, within-group performance variability, convergence time, disagreement, etc.) in their raw distributions as well as their reciprocal relationships, we can collect converging evidence on the mechanisms underlying our main effects, and formulate novel hypotheses. The picture that emerges is a coherent one as we try to outline in the main text and as follows.

Both group size and group diversity affect the distribution of information, forecasts and thus errors in our groups. The advantage of small groups in a complex environment is known to be due to their higher noise with advantages observed in many real life contexts (Kao & Cousin 2014 *PRSB*, Kao & Cousin 2019, *PTRSB*; Wu et al 2019 *Nature*; Galesic et al 2018 *Decision*). The advantage of diverse groups is known to be an informational advantage, but attrition to communication has also been documented (Hong & Page 2004, *PNAS*, Mannix et al 2005). Manipulating these two factors is thus theoretically expected to affect the information of groups. Empirically, and in agreement with contemporary debates on information diversity online, we show that by sorting people who differ on an arbitrary large number of dimensions (the DBSCAN algorithm), we indirectly manipulated the correlation of forecasts between individuals. Importantly this correlation emerges only after people are allowed to search information via their browsers (Results *Multi-dimensional profiling* section). Principal component analysis shows *post-hoc* that few interpretable axes of variations were most important for our manipulation and group composition (Figures S13-23). As expected, diversity and modular-aggregation both reduce the impact of correlated errors at the aggregate level. Importantly however their compound effect (namely in small diverse groups) produces poorer performance in forecasting tasks under time pressure. Small diverse groups consistently showed a disadvantage in a range of metrics. They showed greater brier scores (Figures 2-3), larger variability in performance after online browsing (Figure 4b), larger unresolved disagreement (Figure 4a), longer convergence (Figure S7) and larger mean absolute convergence values (Figure S6).

We are always happy to test alternative hypotheses to explain the same data. However the picture that emerges from converging analyses is coherent with our interpretation. We agree with the reviewer that plotting regression model results (partial residuals) without the raw data, and especially when these models were not precisely pre-registered, is not appropriate. We apologise for our lack of attention to details and thank the reviewer to prompt us to improve the manuscript further. In this new resubmission, we thus introduce plots of raw Brier score distributions as requested by the reviewer.

Raw Brier score distributions. We report below the plots requested by the reviewer. These appear also in the Supplementary material of the manuscript. We introduced a boxen plot representing the distribution of Brier errors, broken down by stage and condition, both at the individual level (roughly corresponding to Figure 2 and Table 2a-b) and at the aggregated level within each group (roughly corresponding to Figure 3 and Table 2c-d). The interaction between diversity condition and modularity can be observed when looking at the central tendencies of the distributions. This, with the addition of the partial residual plots in the main text, and the supplementary figures reported above, should give the reader a full picture of the raw data, main effects, as well as insight into possible underlying mechanisms.

Figure. Upper panel: Distributions of Brier error scores divided by condition and forecasting stage for individual forecasts. Lower panel: Distributions of Brier error scores divided by condition and forecasting stage for within-group aggregated forecasts. Although diversity is beneficial in large groups (low modularity), it damages forecast accuracy in smaller groups (high modularity).

List of relevant edits:

- We report in supplementary material, distributions of raw brier scores (not partial residuals), divided by condition, forecasting stage, both at the individual and aggregated level.
- We remove claims of pre-registration.
- We provide information regarding how to access pre-registered material for the interested reader.
- We explicitly say that we did not pre-register analysis

- We clarify that only Table 2a and 2d and Table S14 were hypothesis driven
- We clearly spell out throughout the manuscript that most analyses were instead carried out post-hoc or with exploratory purposes.
- We add further figures of important indicators of group information processes, like disagreement, and variability in performance within each group.

Re: pre-registration

These clarifications do not go far enough. The statistical tests in the pre-registration list a large number of common statistical test families and provide far too many researcher degrees of freedom. Even with the added caveats, I do not feel it is appropriate to claim the statistical tests were sufficiently pre-registered. In general, they should include which tests/variables/etc.. will be used to test which hypotheses and criteria that will be used for evaluating the results.

AUTHORS: We agree with the reviewer that the fault was on our side in under-specifying analysis and statistical tests (ie. providing broad classes of analysis rather than specific regression models with specific predictors). We cannot unfortunately undo this mistake, but we greatly appreciate the reviewer's point of view as it will make our pre-registration practices more strict in the future. We hope we can convince the reviewer that this mistake was not done with malice, but was a genuine one. As a token of our sincerity, we took the reviewer's recommendation on board and removed all claims of pre-registration (including from the abstract). We clearly say "*Analyses were not pre-registered.*" (Ln 96). We think some readers will still find some value in reading the pre-registration material that we share in the manuscript because of the hypotheses that are spelled out therein. These provide some background information on the motivations behind the study and some of the trends that we were expecting given the literature on the topic.

List of relevant edits:

- Removed strong claims of pre-registration
- Specified that analyses were not pre-registered
- Clarified the preliminary and exploratory nature of the study

Re: stats

There are approximately 60 tests of significance (e.g. p-values) throughout the text including those that are non-significant.

False positive rates and overfit will increase with model complexity. Between this and the researcher degrees of freedom afforded by the pre-registration it remains difficult to disentangle the key findings from statistical artefacts.

AUTHORS: Once again, we thank the reviewer for their comments, as we now understand the meaning of the 60 tests mentioned in the previous revision round. We agree with the reviewer that it is important to simplify the reporting of the analyses and distinguish between main hypothesis-driven analyses and exploratory analyses. In this new submission, we thus simplify the reporting of the analysis, largely reducing redundant passages and avoiding reporting similar effects twice (which contributed to a feeling of multiple comparisons). We also clarify which ones are our main analyses and which are purely exploratory. We outline and explicitly acknowledge the reviewer's criticism in the discussion. We provide below further thoughts that may address this concern.

Main hypotheses. We agree with the reviewer that the statistical tests were under-specified in our pre-registration material. We hope that the fact that the hypotheses under consideration were clearly spelled out may alleviate some of the concerns:

...we expect that aggregating consensus forecasts and final individual forecasts (see below) across small groups will lead to better forecasting accuracy than aggregating within groups or over members of large groups (H1). We expect diverse groups to perform better than homogeneous groups and small group-based aggregation to perform better than large group-based aggregation (H2). We do not have specific expectations on how the two dimensions will interact with each other. We expect that private revised forecasts (see below) will show reduced variability compared to private initial forecasts (H3). Overall we expect a reduction in variability as people interact with each other (see Lorenz et al 2011) (H4) but increase in accuracy due to private evidence gathering (H5) and social processes (H6).

Our emphasis.

Main analysis. We agree with the reviewer that (a) it is important to clearly spell out for the reader which analysis should be considered main analyses and which are exploratory in nature; and (b) the simplest possible models should be used to test one's hypotheses. Regarding (a), this information is clearly spelled out throughout the paper in the new resubmission. Regarding (b), we applied this Occam's razor approach to produce our main results (ie. Table 2, Table S14). We report exact formulas for openness so the reviewer and the editors can confirm that we did not, for example, added unnecessary complexity or *ad hoc* variables. The regression models reflect our original hypotheses very closely. We report below regression models and associated hypotheses. Formulae can also be found in Supplementary material but were removed from the main text for the sake of space and presentation quality. We think these regressions represent the simplest models that we could run to test the hypotheses specified in the pre-registration material that still can account for non-independence of data points. We included Table 2b and 2c to provide a fuller picture but we are happy to remove it if the editor suggests so doing. We clarified in the main text which hypotheses were disconfirmed by the data (e.g. H3).

Table	Hypothesis	Model
Table 2a	H5-H6	individual error ~ forecasting stage + (1 player_id:group_id) + (1 question_id)
Table 2b		individual error ~ forecasting stage + diversity*size + (1 player_id:group_id) + (1 question_id)
Table 2c		group error ~ forecasting stage + (1 question_id)
Table 2d	H1-H2	group error ~ forecasting stage + diversity*size (1 question_id)
Table S14	H3-H4	disagreement~type+diversity*modularity+ (1 question_id)

Table. The table above shows the list of tables found in the main text and supplementary material that should be considered as our main analysis, as opposed to exploratory data analysis. For each table we also show whether it represents a specific hypothesis from our pre-registration material, and the exact expression used to test the hypothesis. Formulas were compared with alternative ones (more or less complex) using standard model selection criteria based on standard information criteria.

AIC selection. Notice in the Table above that although Table 2d in the main text tests H1-H2, there was no mention of forecast type in H1 and H2. We are more than happy to remove this predictor if the editor thinks it removes unnecessary complexity.

We notice that all models' expressions were tested against more complex and simpler ones, and compared using standard objective model selection methods. We used information theoretic methods such as AIC comparisons. This approach should alleviate the type of valid concerns that the reviewer is referring to (ie. overfitting the data). Thus, all models found in our main analyses were not "cherry picked", but selected via standard data-driven or hypothesis-driven model selection procedures. This type of approach is consistent with what we have found in the literature.

Exploratory and post-hoc analysis. We thank the reviewer for clarifying their meaning regarding the number of tests. We understand that the reviewer must have also included tests/p-values in the supplementary material (we counted little more than 30 p-values in the main text). We would like to address this concern and ask the reviewer to consider a few points. First, many p-values and statistics in the main text are reported both in tables as well as in the text, but are in fact identical. We adopted such redundancy to favor clarity in a relatively complex study. Second, we would like to draw attention on the fact that the 60 different tests are not all equal, because they concern very different types of analyses. As mentioned above, and clarified in this new submission throughout the manuscript, most analyses are purely exploratory

and data-driven. Only models that appear in the list above are actually the focus of our hypotheses. We apologize for not being clearer in our previous submissions. We think we have greatly improved the clarity and scope of our paper. Thus, treating all analyses as the same would be incorrect. The reviewer certainly recognizes the important role that exploratory analysis plays in research, insofar it is clearly spelled out to the reader which effects were hypothesis-driven and which effects were found after exploring the data. This is clear by the following anecdote. If researchers had to reduce the significance threshold (e.g. apply a Bonferroni correction) for every added exploratory test, there would be no incentive for researchers to explore their data or if they did, there would be negative incentives in reporting them. This is because for every added exploration, the researcher would need to ‘penalize’ their main hypotheses.

In line with toning down the bold claims of the paper and showing greater humbleness in our interpretation, we apply in this new submission a Bonferroni correction to our main test (Table 2, S14), thus reducing the significance threshold by the number of predictors of the model. We observe that we lose only a few effects, which do not drastically change our main story. E.g., we lose the main effect of diversity and the interaction at the aggregate level but not at the individual level (corrected threshold of 0.005). Please notice however that a Bonferroni correction is highly conservative as it assumes all tests are independent. However, in our study, this is not the case. As seen in Table 2 and the formulas reported above,

- a. Table 2b is an extension of 2a, and Table 2d is an extension to 2c.
- b. The aggregated forecasting error measure is deterministically derived from the individual forecasting error measure (it is the median error of group)

To conclude, we fully agree with the critical reading of the reviewer that the manuscript in its previous form was underspecified and gave the impression of testing too many things and with arbitrary criteria. For this reason, in the new submission, we remove all mentions of pre-registration from the abstract and the remaining text. We provide our pre-registration material to the interested reader so as to inform them on what behaviors we expected before the data was collected. This is particularly useful when the data disconfirmed our hypotheses (e.g., in the case of H3). We clarify throughout the paper which tests were post-hoc and/or exploratory. Importantly, we spend much more attention in providing full disclosure of raw data distributions of various measures considered, including Brier scores (Figure S2), spread of opinions or “disagreement” measure (Figure 4a), and variability in performance (Figure 4b, kindly requested by another reviewer).

List of relevant edits:

- We edited the text to highlight which statistical tests and analyses are exploratory
- Reduced the number of analyses reported, avoided repetitions of similar results, simplified passages and avoided redundancies.
- We clarify that our main results are the ones reported in Table 2 (a,d) and Table S14. They reflect our main hypotheses as closely and parsimoniously as possible (with caveat of Figure 2d as described above)

- We specify in Table 2 the effects that do not survive a multiple-comparison (Bonferroni) correction.
- We specify in the discussion that our study should be interpreted as “preliminary evidence” rather than a conclusive research.

Re: first order vs. second order accuracy

The use of alternate (e.g. logit) link functions is valuable, but note that log link functions are used to model error in numeric estimation tasks is fundamentally different from discrete choice (even with probability) tasks.

the clarification on Brier scores as a dependent variable of interest is appreciated.

AUTHORS:

We are happy that our clarifications were deemed valuable and acknowledge the difference between numeric and discrete choices cases, here and in the main text.

Re: quality of plotting

Adding same-color and difficulty to distinguish dots on top of a dynamite plot doesn't really help with the display of information. Suggest removing the bar entirely and perhaps replacing with a box and jittered-points. This is particularly true as the measure can (and does) go below zero.

AUTHORS:

We thank the reviewer once more because their comment helped us improve a confusing figure. We have opted to provide the full distributional information about within-group conflict (now renamed disagreement after another reviewer's suggestion), broken down by forecast stage, diversity and group size. The new figure conforms to the stylistic requirements of the reviewer. We also include a second panel to the same figure showing the variability in performance within groups, as kindly suggested by another reviewer (Figure 4b).

The addition of a jitter plot on top of the boxen plot unfortunately resulted in overcrowding the figure so we report it here for full disclosure but this was not added to Figure 4a.

Figure. Distribution of disagreement broken down by forecast type, group size and diversity condition. The plot complements Figure 4a. It was not added on top of Figure 4a for consistency with other figures and to preserve presentation quality.

List of relevant edits:

- Replaced all dynamite plots with boxen plots
- Replaced Figure 4 with a more informative multi-panel figure
- Included full distributional information about disagreement (previously “conflict”) across conditions
- Included a second panel to Figure 4, with information regarding the variability in performance within groups (suggested by R1). This factor is known to affect collective performance in a number of domains

In general the plotting aesthetics are still problematic. Some have white backgrounds, some have grey. Some have a y-axis, some don't. Colormaps seem somewhat random, and often have color-blindness issues (e.g. red/green in the same plot).

AUTHORS:

We agree with the reviewer's feedback. We thank the reviewer for their comment and their attention to details because it greatly improved our figures and presentation style. Figures in this resubmission are more consistent with each other, and centrally produced by our department's graphics team. We provide further details below as needed.

The colormap of Figure 1 was made by a contracted graphic designer so that people affected by protanopia or protanomaly (the most common forms of color-blindness) could still notice a difference between sample segments and that this difference reflected the trait difference represented by the distance on the survey space. We used a freely-available online converter (<https://www.color-blindness.com/coblis-color-blindness-simulator/>) to simulate what a person affected by protanopia would see when looking at Figure 1. We show below the converted

figure. As intended, the Core and Inner segments (ie. the people who form the homogeneous groups in our diversity manipulation) are indeed more similar (yet detectably different) to each other than the core and outer segments (ie. people who form the diverse groups in our diversity manipulation). This can also be seen in panel b and c in our manipulation procedure description.

However, to seek further confirmation and to address the reviewer’s concern, we asked for a second opinion from our Department’s graphics team. They confirmed that they “see no issue with the color scheme and color blind people should be able to understand [your] intention”.

We thank the reviewer for raising this valid concern.

Figure. Figure 1 in the main text was converted into a color scale reflecting the distribution of colors seen by people affected by protanopia or protanomaly (the most common forms of color blindness). Notice that (a) characters corresponding to core and inner segments in the figure are more similar to each other than characters corresponding to core and outer segments. This is in line with the Euclidean distance metric that this schematic wants to represent. Nevertheless, characters corresponding to core and inner segments (e.g. panels c) are still distinguishable.

Regarding figure 2, we kept a neutral grey scale as there is no need to distinguish for different segments (all points refer to the core segment). We changed however the axis labels in accordance with the reviewer's suggestions (namely y-axis instead of titles). We think the figure greatly improved as a result of their kind suggestion.

We also agree with the reviewer that figure 4 did not match either Figure 1 or Figures 2 and 3, and represented a stylistic departure from the color-scheme. We have thus adapted it to a grey scale, in accordance with Figures 2 and 3. We have made y-axis and titles more consistent with each other. We also introduced more panels and conditions to reflect other reviewers' suggestions. We believe the figures in this new submission have greatly improved.

List of relevant edits:

- Replaced all dynamite plots with boxen plots
- Removed titles from figures
- Added y-axis labels
- Improved consistency across figures
- Changed the color-scheme of Figure 4 to reflect the one in Figure 2-3
- Added multi-panels to Figure 4
- Provided editable files to the editorial team so to reflect the journal's internal policies in formatting figures and captions

Dynamite plots persist in the SI

AUTHORS:

We apologize for overlooking this plot and thank the reviewer for noticing it. We have changed the plot to a more appropriate boxen plot (also reproduced below). Boxen plots have been used throughout the new version of the manuscript and supplementary material to give better characterization of the full distribution of the measures plotted. This gave us also the opportunity to understand a feature of the data that was overlooked before, namely that small diverse teams in our study were noisier in their convergence process than all other groups, as shown by the larger standard deviation in mean absolute convergence. We stress the novel result in the main text. We have edited the manuscript accordingly to include this result.

Figure. Figure S6 in supplementary material. The figure has been replotted to remove bar plots. Please notice that, coherently with the rest of the analyses in the paper, small diverse groups show greater noise in the convergence measure.

Those giant grids of tiny plots in the SI are still fairly confusing and largely unreadable.

AUTHORS:

The plot was intended to show that the distribution of forecasts across the probability spectrum was not skewed towards the 0% (the event does not happen). We agree it may have been overwhelming and we apologize for it. We removed it and replaced it with the more appropriate standard detection theoretical analysis of bias (criterion c), mentioned above.

Re: the specificity of a task

Consistency with previous methods and constraints do nothing to ameliorate the scientific concerns regarding the oddness of the behavioral task.

AUTHORS:

We are happy to address this concern further. We took steps to address it. We have modified the title, the main focus in the introduction and the interpretation of our results in the discussion to reflect the specificity of our task. We removed any claim at explaining the effects of diversity *tout court* (this was never our intention, and we apologize if our previous wording suggested otherwise) and instead make extra clear that our experiment pertains only to a specific domain, namely complex forecasting tasks under conditions of uncertainty and time sensitivity.

We would also like to offer further points of reflection. We did not come up with the specific forecasting task. As the reviewer acknowledges in one of their previous comments, the forecasting problems were pre-selected by the national intelligence agency for a national forecasting tournament. They were reckoned as representative of real-world high stakes forecasting problems faced by real analysts. We think appreciating this fact can alleviate the

concerns of the reviewer around the choice of the behavioral task. Regarding other sources of concern in our manipulation, the refocusing of the narrative in the new introduction has largely improved the clarity around why we made certain design choices and the value for readers both in the psychology and in the computational social science fields.

Re: Negative density fits

This can be dealt with using the “clip” option in seaborn (which seems to be the plotting library used)

AUTHORS:

We thank the reviewer for this useful technical suggestion. We tried the ‘clip’ parameter. Unfortunately it still did not alleviate the reviewer’s concern as this function still uses the Gaussian kernel method. After plotting the figure with the clip option and finding that it still spilled over in the negative part of the plot, we opted to plot the graph as a boxen plot, thus integrating the benefits of a histogram with the benefits of a density function. The reviewer will notice that boxen plots are now plotted across the manuscript for many of the measures of interest.

Furthermore, the reviewer will notice that we opted to improve the figure and show the full distribution for different conditions and stages of interest. We comment on these new results in this new submission.

** See Nature Research’s author and referees’ website at www.nature.com/authors for information about policies, services and author benefits.

REVIEWER COMMENTS

Reviewer #1 (Remarks to the Author):

I am impressed by your thorough review. I have no further aspects that I would like to highlight. I just noticed that you cited Hong & Page PNAS 2004 and was wondering whether you are aware of the response by Abigail Thompson which seriously questions the results claimed by Hong and Page: Does diversity trump ability? A Thompson - Notices of the AMS, 2014

Reviewer #2 (Remarks to the Author):

The authors have addressed all my comments. Thank you.

Reviewer #3 (Remarks to the Author):

The time taken by the authors to write a full response to each of the points is appreciated, as well as the additional analysis conducted to justify both the measures and methods. Unfortunately there are still substantive issues with the MS that cause me concern:

- 1) The behavioural task remains one of very obscure questions for which it feels difficult to believe there would be substantial signal to study something akin to wisdom or forecasting in any other context.
- 2) While it is a necessary and unfortunate constraint of the study, those classes are highly unbalanced in plausibility with only one being true. Wisdom is indistinguishable from simply measuring the probability of answering “no”.
- 3) The PCA seems to reveal that diversity is mostly driven by race/ethnicity and political orientation. Given the obscurity of the questions, it seems perhaps unsurprising that increasing diversity would increase the chance that at least one of the participants in a group can recall what a “loya jirga” is enough to make a reasonable guess. This effect would be more pronounced in a large group than a small group.

To illustrate these problems, imagine I ask a group of scientists questions along the lines of:

“Is *Campephilus principalis* likely to become an invasive species throughout Australia in the next 20 years?”

If we select a discipline at random, and then make large or small groups they would be unlikely to know what *Campephilus principalis* is and would guess True with some probability > 0 .

Now, if we compose groups of scientist randomly chosen across disciplines, a small group doesn't do much better than a group from a single discipline because the odds of containing an ornithologist remains low.

However, the odds of happening upon an ornithologist increase nonlinearly with group-size and a finite number of academic disciplines. If there happens to be an ornithologist, they can trivially identify the answer to this question as "No" because Ivory-Billed Woodpeckers are largely believed to be extinct were last seen in the US.

Imagine we have a set of a few questions from across a large range of disciplines, all of which are obviously unlikely to anyone with domain-knowledge. Diversity would seem to improve forecasting in large, but not small groups. Large groups are more likely to get a single question or two correct because they have an expert.

Critically, because the probability of the events is low, brier error will be high in anyone without domain knowledge that assumes the events have closer to equal probability of occurring.

Also as noted by another referee also the way distance is measured is problematic, and still think that the diversity and the topic are very specific and thus the title misrepresents what we actually can take from this regarding general principles.

RESPONSE TO REVIEWER COMMENTS

Reviewer #1 (Remarks to the Author):

I am impressed by your thorough review. I have no further aspects that I would like to highlight. I just noticed that you cited Hong & Page PNAS 2004 and was wondering whether you are aware of the response by Abigail Thompson which seriously questions the results claimed by Hong and Page:

Does diversity trump ability? A Thompson - Notices of the AMS, 2014

We were very happy with the positive feedback of the reviewer. We were not aware of this paper, and were happy to remove the Hong&Page paper from our manuscript.

Reviewer #2 (Remarks to the Author):

The authors have addressed all my comments. Thank you.

We were pleased with the reviewer's positive assessment of our work.

Reviewer #3 (Remarks to the Author):

The time taken by the authors to write a full response to each of the points is appreciated, as well as the additional analysis conducted to justify both the measures and methods.

Unfortunately there are still substantive issues with the MS that cause me concern:

We appreciate the reviewer's positive evaluation of our last resubmission, and are happy to provide further clarifications.

1) The behavioural task remains one of very obscure questions for which it feels difficult to believe there would be substantial signal to study something akin to wisdom or forecasting in any other context.

We appreciate the reviewer's point of view, and agree that the previous submission did not discuss this issue deeply enough. We offer two important reasons supporting our design choice and the selection of these specific questions, both below and in the revised manuscript.

First, we argue that these questions are representative of the specific domain we are interested in investigating, namely geo-political forecasting. These are forecasting problems that were pre-selected by a national forecasting tournament aimed at improving national security's geo-political forecasting capacity. The fact that they were not 'cherry-picked' by the authors will reassure the reader. Problems were selected by a national agency to be a **representative sample** of intelligence problems commonly tackled by intelligence analysts, every day.

Suggesting that these questions are too obscure to contain any signal to study wisdom/forecasting ability seems a bit extreme to us. Perhaps we misunderstood the reviewer's comment, and we would be glad if they could provide some clarifications in this regard. Does the reviewer think that the questions are uninformative to study geo-political forecasting in general, or that they are useful to study geo-political forecasting, but this does not generalize to other collective intelligence tasks? We agree with the latter interpretation and we have clearly

stated the limitations of our results throughout the paper, starting from the title. However, if the reviewer thinks the former, we must politely disagree. The reviewer's position suggests that IARPA's questions were not representative samples as IARPA claims. We believe that a debate over the validity of stimuli selected by a national forecasting competition is beyond the scope of this response and we let the editor make a final judgment on this matter. As experimenters, we simply assumed IARPA's claims were valid, and took a random subsample of IARPA provided questions that were still unresolved at the time.

The second point that we hope the reviewer can appreciate is the difficulty and specificity of the questions also served a very precise methodological purpose. We wanted group discussions in the experiment to be driven (possibly entirely driven) by the information participants could retrieve online during the time allotted by the experiment, rather than domain-specific information already possessed by the participant before the experiment. This is very important because, as we have made clear in the text, we were especially interested in studying the ability of groups to **forage information online using technology** (search engines in this case). This is made very clear in the first paragraph of the manuscript, which reads: *'One question is whether recommendation algorithms and homophily can impact the ability of online groups to collectively search and use information to form accurate predictions, especially under high time pressure and uncertainty'*. Less obscure questions—for example sport betting (What are the chances that Italy will win the next Football World Cup?), or geo-political events closer to Western culture ("What are the chances that Donald Trump will be re-elected in 2020?")— would have not been relevant for answering our scientific question. This is because participants' judgment would have been highly biased towards what they knew already. The setup suggested by the reviewer, containing perhaps difficult but 'common' forecasting problems, would have been perfect to answer the scientific question: How does group composition influence the aggregation of different opinions? This is not a novel question in our opinion, as it has been investigated in my many papers in the field. For this reason we believe that the setup suggested by the reviewer would have not been able to answer whether group composition affects the group's ability to forage domain-specific information online to form accurate representations of future events. For these reasons, and all the others listed in the previous two rounds of review, we believe our setup is perfectly suited for the scope of our investigation.

The reviewer had suggested in the previous round of revision to change the title to highlight the specificity of the task, saying "the title would be better worded as 'forecasting geopolitical events'" Accordingly, we were happy to change the title to "Multi-trait diversity improves geo-political forecasting accuracy in large but not small online groups", thus adding the qualifier that the reviewer suggested.

In the introduction, we clearly lay out the importance for us to investigate the impact of common digital technologies (search engines and news recommendation algorithms in this case) on collective forecasting judgements, rather than simply group composition. We have downplayed the generality of our claims regarding collective intelligence. We certainly do not believe nor state that these effects apply to all group interactions (e.g., online and offline, face to face and via text, etc.), or with different operationalizations of the constructs under investigation. In the

discussion and throughout the paper, we argue that this behavioral task is needed to answer the specific scientific question under consideration in this paper (Ln 478-498).

The debate around the representativeness of questions, the task selection methods and other considerations are also explained in more detail in the text.

2) While it is a necessary and unfortunate constraint of the study, those classes are highly unbalanced in plausibility with only one being true. Wisdom is indistinguishable from simply measuring the probability of answering “no”.

We understand the point made by the reviewer and believe their concern needs to be discussed more thoroughly in the paper. What is not clear in this alternative interpretation of the results, is the psychological and behavioral mechanism behind it. As clearly shown by our bias analysis (Figure S9), people did not show an initial bias towards answering “no”. If anything, this phenomenon emerged in later stages of the experiment. But to the best of our knowledge, there is no plausible explanation for why social interaction would lead to a unilateral increase in the chance of answering “no”. In fact, there is evidence that this should *not* be the case. Please notice that it is known that social interaction tends to extremize initially held opinions, a phenomenon popularly known as polarization or ‘risky shift’ (Moscovici 1969). Given that opinions were initially balanced before interactions around 50% (as shown by our bias analysis, Figure S9), one should expect that social interactions lead groups to gravitate towards 0 and 100% symmetrically. The criticism offered by the reviewer (that wisdom is indistinguishable from people answering NO after social interaction) is understandable, but not supported by our bias analysis, nor by known phenomena in social psychology. Importantly, the reviewer has not yet clarified how such unspecific bias towards lowering probability could generate the specific interaction observed between group size and diversity. Without an explanation of the mechanisms underlying this potential confound, we would feel uncomfortable arguing in the paper for this explanation. We thus need to conclude that groups underwent a process of information aggregation via deliberation that extrapolated a weak signal gathered online. This position is more in line with past literature on group deliberation and social learning. This explanation is further corroborated by our manual labeling of conversations. As described in the discussion, we have manually labelled each conversation and counted the number of individuals who seemed to possess some degree of knowledge about the topics under consideration. About half people in each group showed some degree of domain-knowledge. This suggests that discussions were based on evidence and opinions rather than an unspecific or unmotivated runaway towards the lower end of the probability scale.

We have edited our discussion on this possible confound and clarified pros and cons of this interpretation at length. (Ln 498-530)

3) The PCA seems to reveal that diversity is mostly driven by race/ethnicity and political orientation. Given the obscurity of the questions, it seems perhaps unsurprising that increasing diversity would increase the chance that at least one of the participants in a group can recall what a “loya jirga” is enough to make a reasonable guess. This effect would be more pronounced in a large group than a small group.

To illustrate these problems, imagine I ask a group of scientists questions along the lines of:

“Is *Campephilus principalis* likely to become an invasive species throughout Australia in the next 20 years?”

If we select a discipline at random, and then make large or small groups they would be unlikely to know what *Campephilus principalis* is and would guess True with some probability > 0 .

Now, if we compose groups of scientists randomly chosen across disciplines, a small group doesn't do much better than a group from a single discipline because the odds of containing an ornithologist remains low.

However, the odds of happening upon an ornithologist increase nonlinearly with group-size and a finite number of academic disciplines. If there happens to be an ornithologist, they can trivially identify the answer to this question as “No” because Ivory-Billed Woodpeckers are largely believed to be extinct were last seen in the US.

Imagine we have a set of a few questions from across a large range of disciplines, all of which are obviously unlikely to anyone with domain-knowledge. Diversity would seem to improve forecasting in large, but not small groups. Large groups are more likely to get a single question or two correct because they have an expert.

Critically, because the probability of the events is low, brier error will be high in anyone without domain knowledge that assumes the events have closer to equal probability of occurring.

Also as noted by another referee also the way distance is measured is problematic, and still think that the diversity and the topic are very specific and thus the title misrepresents what we actually can take from this regarding general principles.

We really appreciate this insightful perspective offered by the reviewer and were very happy to include it in our discussion regarding the mechanism underlying the effects under study. Their example clearly shows the reviewer's statistical argument. We do agree that diversity so measured (ie. population variance across socio-cultural and political dimensions) would have had a beneficial effect on large groups (more likely to contain an expert) than in small groups (less likely to contain an expert). Importantly however, we point out in our revised discussion that one result that this explanation does not account for is the fact that (at least at the aggregate level) interacting in small homogeneous groups was beneficial for individual forecast accuracy, as well as for aggregate forecasts. In other words it does not explain why we observed the reversed effect in homogeneous groups, nor why aggregating small homogeneous groups forecasts gave the best result.

Furthermore, this explanation predicts that larger performance variability will be observed in large diverse groups because these are the ones that are more likely to contain a majority of uninformed guessers and an informed minority of experts. However, this was not the case in our study, where large diverse groups showed lower performance variability in stage 1 and

performance variability equivalent to large homogeneous groups in following stages (see Figure 4b). It seems instead that the largest performance variability was observed in small diverse groups, and specifically after they retrieved information online. These findings should clarify that although the logic described by the reviewer is certainly one aspect, technological (individuals interacting with their search engines) and social aspects (individuals interacting with others) are also an important part of the story.

We have reviewed our discussion and introduced the reviewer's poignant example to make their insightful point. Furthermore, we are happy to change the title once more (perhaps "Socio-cultural diversity improves geo-political forecasting accuracy in large but not small online groups"?) if the editor deems this necessary. However, please notice that we did not manipulate "socio-cultural diversity" but indeed "multi-trait diversity".

REVIEWERS' COMMENTS

Reviewer #3 (Remarks to the Author):

I am happy with the Author's responses.